# IMPROVING MULTI-TASK LEARNING VIA SEEKING TASK-BASED FLAT REGIONS

## ABSTRACT

Multi-Task Learning (MTL) is a widely used and powerful learning paradigm for training deep neural networks that allows learning more than one objective by a single backbone. Compared to training tasks separately, MTL significantly reduces computational costs, improves data efficiency, and potentially enhances model performance by leveraging knowledge across tasks. Hence, it has been adopted in a variety of applications, ranging from computer vision to natural language processing and speech recognition. Among them, there is an emerging line of work in MTL that focuses on manipulating the task gradient to derive an ultimate gradient descent direction to benefit all tasks. Despite achieving impressive results on many benchmarks, directly applying these approaches without using appropriate regularization techniques might lead to suboptimal solutions to real-world problems. In particular, standard training that minimizes the empirical loss on the training data can easily suffer from overfitting to low-resource tasks or be spoiled by noisy-labeled ones, which can cause negative transfer between tasks and overall performance drop. To alleviate such problems, we propose to leverage a recently introduced training method, named Sharpness-aware Minimization, which can enhance model generalization ability on single-task learning. Accordingly, we present a novel MTL training methodology, encouraging the model to find task-based flat minima for coherently improving its generalization capability on all tasks. Finally, we conduct comprehensive experiments on a variety of applications to demonstrate the merit of our proposed approach to existing gradient-based MTL methods, as suggested by our developed theory. Our training code is available at https://github.com/anonymous-user00/FS-MTL.

## 1 INTRODUCTION

Over the last few years, deep learning has emerged as a powerful tool for functional approximation by exhibiting superior performance and even exceeding human ability on a wide range of applications. In spite of the appealing performance, training massive independent neural networks to handle individual tasks requires not only expensive computational and storage resources but also long runtime. Therefore, multi-task learning is a more preferable approach in many situations (Zhang et al., 2014; Liu et al., 2019a; Wang et al., 2020) as they can: (i) avoid redundant features calculation for each task through their inherently shared architecture; and (ii) reduce the number of total trainable parameters by hard parameter sharing (Kokkinos, 2017; Heuer et al., 2021) or soft parameter sharing (Gao et al., 2019; Ruder et al., 2019). However, existing state-of-the-art methods following the veins of gradient-based multi-task learning (Sener & Koltun, 2018; Yu et al., 2020; Liu et al., 2021a; 2020; Javaloy & Valera, 2021; Navon et al., 2022) tend to neglect geometrical properties of the loss landscape yet solely focus on minimizing the empirical error in the optimization process, which can be easily prone to the overfitting problem (Kaddour et al., 2022; Zhao et al., 2022).

Meanwhile, the overfitting problem of modern neural networks is often attributed to high-dimensional and non-convex loss functions, which result in complex loss landscapes containing multiple local optima. Hence, understanding the loss surface plays a crucial role in training robust models, and developing flat minimizers remains one of the most effective approaches (Keskar et al., 2017b; Kaddour et al., 2022; Li et al., 2022; Lyu et al., 2022). To be more specific, recent studies (He et al., 2019; Zheng et al., 2021) show that the obtained loss landscape from directly minimizing the empirical risk can consist of many sharp minimums, thus yielding poor generalization capacity

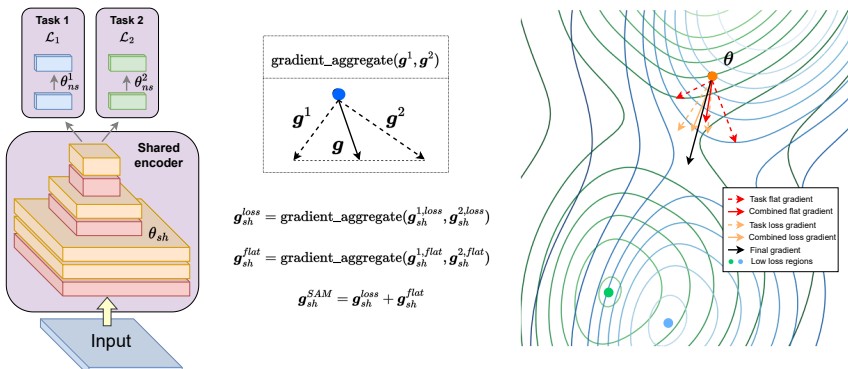

Figure 1: We demonstrate our framework in a 2-task problem. For the shared part, *task-based flat gradients* (red dashed arrows) steer the model to escape sharp areas, while *task-based loss gradients* (orange dashed arrows) lead the model into their corresponding low-loss regions. In our method, we aggregate them to find the combined flat gradient $\boldsymbol{g}_{sh}^{flat}$ and combined loss gradient $\boldsymbol{g}_{sh}^{loss}$, respectively. Finally, we add those two output gradients to target the joint low-loss and flat regions across the tasks. Conversely, updating task-specific non-shared parts is straightforward and much easier since there is the involvement of one objective only.

when being exposed to unseen data. Moreover, this issue is apparently exacerbated in optimizing multiple objectives simultaneously, as in the context of multi-task learning. Certainly, sharp minima of each constituent objective might appear at different locations, which potentially results in large generalization errors on the associated task. To this end, finding a common flat and low-loss valued region for all tasks is desirable for improving the current methods of multi-task learning.

**Contribution.** To further address the above desideratum, we propose a novel MTL training method, incorporating the recently introduced optimization sharpness-aware minimization (SAM) (Foret et al., 2021) into existing gradient manipulation strategies in multi-task learning to further boost their performance. Guiding by the generalization error in Theorem 1, the proposed approach not only orients the model to the joint low empirical loss value across tasks but also encourages the model to reach the task-based flat regions. Importantly, our approach is model-agnostic and compatible with current gradient-based MTL approaches (see Figure 1 for the overview of our approach). By using our proposed framework, the gradient conflict across tasks is mitigated significantly, which is the goal of recent gradient-based MTL studies in alleviating negative transfer between tasks. Finally, we conduct comprehensive experiments on a variety of applications to demonstrate the merit of our approach for improving not only task performance but also model robustness and calibration. Last but not least, to the best of our knowledge, ours is the first work to improve multi-task learning by investigating the geometrical properties of the model loss landscape.

## 2 RELATED WORK

### 2.1 MULTI-TASK LEARNING

In multi-task learning (MTL), we often aim to jointly train one model to tackle multiple different but correlated tasks. It has been proven in prior work (Caruana, 1997; Liu et al., 2019a;b; Ruder, 2017) that it is not only able to enhance the overall performance but also reduce the memory footprint and fasten the inference process. Previous studies on MTL often employ a hard parameter-sharing mechanism along with light-weight task-specific modules to handle multiple tasks.

**Pareto multi-task learning.** Originated from Multiple-gradient descent algorithm (MGDA), a popular line of gradient-based MTL methods aim to find Pareto stationary solutions, from which we can not further improve model performance on any particular task without diminishing another (Sener & Koltun, 2018). Moreover, recent studies suggest exploring the whole Pareto front by learning diverse solutions (Lin et al., 2019; Liu et al., 2021b; Mahapatra & Rajan, 2020; 2021), or profiling the entire Pareto front with hyper-network (Lin et al., 2020; Navon et al., 2021). While these methods are theoretically grounded and guaranteed to converge to Pareto-stationary points, the experimental results are often limited and lack comparisons under practical settings.

**Loss and gradient balancing.** Another branch of preliminary work in MTL capitalizes on the idea of dynamically reweighting loss functions based on gradient magnitudes (Chen et al., 2018), task homoscedastic uncertainty (Kendall et al., 2018), or difficulty prioritization (Guo et al., 2018) to balance the gradients across tasks. More recently, PCGrad (Yu et al., 2020) developed a gradient manipulation procedure to avoid conflicts among tasks by projecting random task gradients on the normal plane of the other. Similarly, (Liu et al., 2021a) proposes a provably convergent method to minimize the average loss, and (Liu et al., 2020) calculates loss scaling coefficients such that the combined gradient has equal-length projections onto individual task gradients.

## 2.2 Flat minima

Flat minimizer has been found to improve generalization ability of neural networks because it enables models to find wider local minima, by which they will be more robust against shifts between train and test losses (Jiang et al., 2020; Petzka et al., 2021; Dziugaite & Roy, 2017). This relationship between generalization ability and the width of minima is theoretically and empirically studied in many studies (Hochreiter & Schmidhuber, 1994; Neyshabur et al., 2017; Dinh et al., 2017; Fort & Ganguli, 2019), and subsequently, a variety of methods seeking flat minima have been proposed (Pereyra et al., 2017; Chaudhari et al., 2017; Keskar et al., 2017a; Izmailov et al., 2018). For example, (Keskar et al., 2017a; Jastrzebski et al., 2017; Wei et al., 2020) analyze the impacts of different training factors, such as batch-size, learning rate, covariance of gradient, dropout, on the flatness of found minima. Additionally, several schemes pursue wide local minima by adding regularization terms to the loss function (Pereyra et al., 2017; Zhang et al., 2018; 2019; Chaudhari et al., 2017), e.g., softmax output's low entropy penalty, (Pereyra et al., 2017), distillation losses (Zhang et al., 2018; 2019).

Recently, SAM (Foret et al., 2021), which seeks flat regions by explicitly minimizing the worst-case loss around the current model, has received significant attention due to its effectiveness and scalability compared to previous methods. Particularly, it has been exploited in a variety of tasks and domains (Cha et al., 2021; Abbas et al., 2022; Qu et al., 2022; Caldarola et al., 2022; Bahri et al., 2022; Chen et al., 2021; Nguyen et al., 2023). A notable example is the improvement that SAM brings to meta-learning bi-level optimization in (Abbas et al., 2022). Another application of SAM is in federated learning (FL) (Qu et al., 2022) in which the authors achieved tighter convergence rates than existing FL works, and proposed a generalization bound for the global model. In addition, SAM shows its generalization ability in vision models (Chen et al., 2021), language models (Bahri et al., 2022) and domain generalization (Cha et al., 2021). However, existing studies have only focused on single task problems. In this work, we leverage SAM's principle to develop theory and devise practical methods, allowing seeking flat minima in gradient-based multi-task learning models.

## 3 Sharpness aware minimization training

Conventional training methods that focus on minimizing the empirical loss can be easily prone to overfitting problems (i.e., the validation error no longer decreases, but the training loss keeps declining), thus, restricting model generalization performance. In an attempt to alleviate such phenomenons, Foret et al. (2021) proposed to minimize the worst-case loss in a neighborhood of the current model parameter given by:

$$\min_{\boldsymbol{\theta}} \max_{||\boldsymbol{\epsilon}||_2 \le \rho} \mathcal{L}\left(\boldsymbol{\theta} + \boldsymbol{\epsilon}\right), \tag{1}$$

where $|| \cdot ||_2$ denotes the $l_2$ norm and $\rho$ represents the radius of the neighborhood. We assume $\mathcal{L}$ is differentiable up to the first order with respect to $\boldsymbol{\theta}$. The optimization problem equation 1 is referred to as *sharpness aware minimization* (SAM).

To solve problem equation 1, Foret et al. (2021) proposed to first find the solution for the inner maximization by approximating $\mathcal{L}(\boldsymbol{\theta} + \boldsymbol{\epsilon})$ via a first-order Taylor expansion w.r.t $\boldsymbol{\epsilon}$ around 0, which is as follows:

$$\boldsymbol{\epsilon}^* = \arg\max_{||\boldsymbol{\epsilon}||_2 \le \rho} \mathcal{L}(\boldsymbol{\theta} + \boldsymbol{\epsilon}) \approx \arg\max_{||\boldsymbol{\epsilon}||_2 \le \rho} \boldsymbol{\epsilon}^\top \nabla_{\boldsymbol{\theta}} \mathcal{L}(\boldsymbol{\theta}) \approx \rho \frac{\nabla_{\boldsymbol{\theta}} \mathcal{L}(\boldsymbol{\theta})}{||\nabla_{\boldsymbol{\theta}} \mathcal{L}(\boldsymbol{\theta})||_2}.$$

Putting into words, the worst-case perturbation is approximated as the scaled gradient of the loss w.r.t the current parameter $\boldsymbol{\theta}$. Then, the gradient w.r.t this perturbed model is computed to update $\boldsymbol{\theta}$:

$$\boldsymbol{g}^{\text{SAM}} := \nabla_{\boldsymbol{\theta}} \max_{||\boldsymbol{\epsilon}||_2 \le \rho} \mathcal{L}\left(\boldsymbol{\theta} + \boldsymbol{\epsilon}\right) \approx \nabla_{\boldsymbol{\theta}} \mathcal{L}(\boldsymbol{\theta} + \boldsymbol{\epsilon})|_{\boldsymbol{\theta} + \boldsymbol{\epsilon}^*} \tag{2}$$

## 4 OUR PROPOSED FRAMEWORK

This section describes our proposed framework to improve existing methods on gradient-based MTL. We first recall the goal of multi-task learning, then establish the upper bounds for the general loss of each task. Subsequently, we rely on these upper bounds to devise the proposed framework for improving the model generalization ability by guiding it to a flatter region of each task.

### 4.1 MULTI-TASK LEARNING SETTING

In multi-task learning, we are given a data-label distribution $\mathcal{D}$ from which we can sample a training set $\mathcal{S} = \{(\boldsymbol{x}_i, y_i^1, ..., y_i^m)_{i=1}^n\}$, where $\boldsymbol{x}_i$ is a data example and $y_i^1, ..., y_i^m$ are the labels of the tasks $1, 2, ..., m$ respectively.

The model for each task $\boldsymbol{\theta}^i = [\boldsymbol{\theta}_{sh}, \boldsymbol{\theta}_{ns}^i]$ consists of the shared part $\boldsymbol{\theta}_{sh}$ and the individual non-shared part $\boldsymbol{\theta}_{ns}^i$. We denote the general loss for task $i$ as $\mathcal{L}_{\mathcal{D}}^i(\boldsymbol{\theta}^i)$, while its empirical loss over the training set $\mathcal{S}$ as $\mathcal{L}_{\mathcal{S}}^i(\boldsymbol{\theta}^i)$. Existing works in MTL, typically MGDA (Sener & Koltun, 2018), PCGrad (Yu et al., 2020), CAGrad (Liu et al., 2021a), and IMTL (Liu et al., 2020), aim to find a model that simultaneously minimizes the empirical losses for all tasks:

$$\min_{\boldsymbol{\theta}_{sh}, \boldsymbol{\theta}_{ns}^{1:m}} \left[ \mathcal{L}_{\mathcal{S}}^1\left(\boldsymbol{\theta}^1\right), ..., \mathcal{L}_{\mathcal{S}}^m\left(\boldsymbol{\theta}^m\right) \right], \quad (3)$$

by calculating gradient $\boldsymbol{g}^i$ for i-th task ($i \in [m]$). The current model parameter is then updated by the unified gradient $\boldsymbol{g} = \text{gradient\_aggregate}(\boldsymbol{g}^1, \boldsymbol{g}^2, \ldots, \boldsymbol{g}^m)$, with the generic operation gradient\_aggregate is to combine multiple task gradients, as proposed in gradient-based MTL studies.

Additionally, prior works only focus on minimizing the empirical losses and do not concern the general losses which combat overfitting. Inspired by SAM (Foret et al., 2021), it is desirable to develop sharpness-aware MTL approaches wherein the task models simultaneously seek low loss and flat regions. However, this is challenging since we have multiple objective functions in (3) and each task model consists of a shared and an individual non-shared parts. To address the above challenge, in Theorem 1, we develop upper bounds for the task general losses in the context of MTL which signifies the concepts of sharpness for the shared part and non-shared parts and then rely on these new concepts to devise a novel MTL framework via seeking the task-based flat regions.

### 4.2 THEORETICAL DEVELOPMENT

We first state our main theorem that bounds the generalization performance of individual tasks by the empirical error on the training set:

**Theorem 1.** *(Informally stated) For any perturbation radius $\rho_{sh}, \rho_{ns} > 0$, under some mild assumptions, with probability $1 - \delta$ (over the choice of training set $\mathcal{S} \sim \mathcal{D}$) we obtain*

$$\left[\mathcal{L}_{\mathcal{D}}^i\left(\boldsymbol{\theta}^i\right)\right]_{i=1}^m \leq \max_{\|\boldsymbol{\epsilon}_{sh}\|_2 \leq \rho_{sh}} \left[\max_{\|\boldsymbol{\epsilon}_{ns}^i\|_2 \leq \rho_{ns}} \mathcal{L}_{\mathcal{S}}^i\left(\boldsymbol{\theta}_{sh} + \boldsymbol{\epsilon}_{sh}, \boldsymbol{\theta}_{ns}^i + \boldsymbol{\epsilon}_{ns}^i\right) + f^i\left(\|\boldsymbol{\theta}^i\|_2^2\right)\right]_{i=1}^m, \quad (4)$$

*where $f^i : \mathbb{R}_+ \to \mathbb{R}_+, i \in [m]$ are strictly increasing functions.*

Theorem 1 establishes the connection between the generalization error of each task with its empirical training error via worst-case perturbation on the parameter space. The formally stated theorem and proof are provided in the appendix. Here we note that the worst-case shared perturbation $\boldsymbol{\epsilon}_{sh}$ is commonly learned for all tasks, while the worst-case non-shared perturbation $\boldsymbol{\epsilon}_{ns}^i$ is tailored for each task $i$. Theorem 1 directly hints us an initial and direct approach.

Additionally, Foret et al. (2021) invokes the McAllester (1999) PAC-Bayesian generalization bound , hence is only applicable to the 0-1 loss in the binary classification setting. In terms of the theory contribution, we employ a more general PAC-Bayesian generalization bound (Alquier et al., 2016) to tackle more general losses in MTL. Moreover, our theory development requires us to handle multiple objectives, each of which consists of the non-shared and shared parts, which is certainly non-trivial.

### 4.3 INITIAL AND DIRECT APPROACH

A straight-forward approach guided by Theorem 1 is to find the non-shared perturbations $\boldsymbol{\epsilon}_{ns}^i, i \in [m]$ independently for the non-shared parts and a common shared perturbation for the shared part. Driven by this theoretical guidance, we propose the following updates.

**Update the non-shared parts.** Based on the upper bounds in Theorem 1, because the non-shared perturbations $\boldsymbol{\epsilon}_{ns}^i, i \in [m]$ are independent to each task, for task $i$, we update its non-shared part $\boldsymbol{\theta}_{ns}^i$:

$$\boldsymbol{\epsilon}_{ns}^i = \rho_{ns} \frac{\nabla_{\boldsymbol{\theta}_{ns}^i} \mathcal{L}_{\mathcal{S}}^i \left( \boldsymbol{\theta}_{sh}, \boldsymbol{\theta}_{ns}^i \right)}{\| \nabla_{\boldsymbol{\theta}_{ns}^i} \mathcal{L}_{\mathcal{S}}^i \left( \boldsymbol{\theta}_{sh}, \boldsymbol{\theta}_{ns}^i \right) \|_2},$$

$$\boldsymbol{g}_{ns}^{i,\text{SAM}} = \nabla_{\boldsymbol{\theta}_{ns}^i} \mathcal{L}_{\mathcal{S}}^i \left( \boldsymbol{\theta}_{sh}, \boldsymbol{\theta}_{ns}^i + \boldsymbol{\epsilon}_{ns}^i \right),$$

$$\boldsymbol{\theta}_{ns}^i = \boldsymbol{\theta}_{ns}^i - \eta \boldsymbol{g}_{ns}^{i,\text{SAM}}, \qquad \text{where } \eta > 0 \text{ is the learning rate.} \tag{5}$$

**Update the shared part.** Updating the shared part $\boldsymbol{\theta}_{sh}$ is more challenging because its worst-cased perturbation $\boldsymbol{\epsilon}_{sh}$ is shared among the tasks. To derive how to update $\boldsymbol{\theta}_{sh}$ w.r.t. all tasks, we first discuss the case when we update this w.r.t. task $i$ without caring about other tasks. Specifically, this task's SAM shared gradient is computed as:

$$\boldsymbol{\epsilon}_{sh}^i = \rho_{sh} \frac{\nabla_{\boldsymbol{\theta}_{sh}} \mathcal{L}_{\mathcal{S}}^i \left( \boldsymbol{\theta}_{sh}, \boldsymbol{\theta}_{ns}^i \right)}{\| \nabla_{\boldsymbol{\theta}_{sh}} \mathcal{L}_{\mathcal{S}}^i \left( \boldsymbol{\theta}_{sh}, \boldsymbol{\theta}_{ns}^i \right) \|_2},$$

$$\boldsymbol{g}_{sh}^{i,\text{SAM}} = \nabla_{\boldsymbol{\theta}_{sh}} \mathcal{L}_{\mathcal{S}}^i \left( \boldsymbol{\theta}_{sh} + \boldsymbol{\epsilon}_{sh}^i, \boldsymbol{\theta}_{ns}^i \right),$$

then we have a straight-forward updating strategy:

$$\boldsymbol{g}_{sh}^{\text{SAM}} = \text{gradient\_aggregate}(\boldsymbol{g}_{sh}^{1,\text{SAM}}, \dots, \boldsymbol{g}_{sh}^{m,\text{SAM}}),$$

$$\boldsymbol{\theta}_{sh} = \boldsymbol{\theta}_{sh} - \eta \boldsymbol{g}_{sh}^{\text{SAM}}.$$

According to our analysis in Section 4.4, each $\boldsymbol{g}_{sh}^{i,\text{SAM}} = \boldsymbol{g}_{sh}^{i,\text{loss}} + \boldsymbol{g}_{sh}^{i,\text{flat}}$ is constituted by two components: (i) $\boldsymbol{g}_{sh}^{i,\text{loss}}$ to navigate to the task low-loss region and (ii) $\boldsymbol{g}_{sh}^{i,\text{flat}}$ to navigate to the task-based flat region. However, a direct gradient aggregation of $\boldsymbol{g}_{sh}^{i,\text{SAM}}, i \in [m]$ can be negatively affected by the gradient cancelation or conflict because it aims to combine many individual elements with different objectives. In this paper, we go beyond this initial approach by deriving an updating formula to decompose SAM gradient into two components, each serving its own purpose, and then combining their corresponding task gradients simultaneously. We also compare our method against the naive approach in Section 5.3.

### 4.4 Our proposed approach

The non-shared parts are updated normally as in Equation (5). It is more crucial to investigate how to update the shared part more efficiently. To better understand the SAM's gradients, we analyze their characteristics by deriving them as follows:

$$\boldsymbol{g}_{sh}^{i,\text{SAM}} = \nabla_{\boldsymbol{\theta}_{sh}} \mathcal{L}_{\mathcal{S}}^i \left( \boldsymbol{\theta}_{sh} + \boldsymbol{\epsilon}_{sh}^i, \boldsymbol{\theta}_{ns}^i \right) \overset{(1)}{\approx} \nabla_{\boldsymbol{\theta}_{sh}} \left[ \mathcal{L}_{\mathcal{S}}^i \left( \boldsymbol{\theta}_{sh}, \boldsymbol{\theta}_{ns}^i \right) \right] + \left\langle \boldsymbol{\epsilon}_{sh}^i, \nabla_{\boldsymbol{\theta}_{sh}} \mathcal{L}_{\mathcal{S}}^i \left( \boldsymbol{\theta}_{sh}, \boldsymbol{\theta}_{ns}^i \right) \right\rangle$$

$$= \nabla_{\boldsymbol{\theta}_{sh}} \left[ \mathcal{L}_{\mathcal{S}}^i \left( \boldsymbol{\theta}_{sh}, \boldsymbol{\theta}_{ns}^i \right) + \rho_{sh} \left\langle \frac{\nabla_{\boldsymbol{\theta}_{sh}} \mathcal{L}_{\mathcal{S}}^i \left( \boldsymbol{\theta}_{sh}, \boldsymbol{\theta}_{ns}^i \right)}{\| \nabla_{\boldsymbol{\theta}_{sh}} \mathcal{L}_{\mathcal{S}}^i \left( \boldsymbol{\theta}_{sh}, \boldsymbol{\theta}_{ns}^i \right) \|_2}, \nabla_{\boldsymbol{\theta}_{sh}} \mathcal{L}_{\mathcal{S}}^i \left( \boldsymbol{\theta}_{sh}, \boldsymbol{\theta}_{ns}^i \right) \right\rangle \right]$$

$$= \nabla_{\boldsymbol{\theta}_{sh}} \left[ \mathcal{L}_{\mathcal{S}}^i \left( \boldsymbol{\theta}_{sh}, \boldsymbol{\theta}_{ns}^i \right) + \rho_{sh} \| \nabla_{\boldsymbol{\theta}_{sh}} \mathcal{L}_{\mathcal{S}}^i \left( \boldsymbol{\theta}_{sh}, \boldsymbol{\theta}_{ns}^i \right) \|_2 \right] \tag{6}$$

where in $\overset{(1)}{\approx}$, we apply the first-order Taylor expansion and $\langle \cdot, \cdot \rangle$ represents the dot product.

It is obvious that following the negative direction of $\boldsymbol{g}_{sh}^{i,\text{SAM}}$ will minimize the loss $\mathcal{L}_{\mathcal{S}}^i \left( \boldsymbol{\theta}_{sh}, \boldsymbol{\theta}_{ns}^i \right)$ and the gradient norm $\| \nabla_{\boldsymbol{\theta}_{sh}} \mathcal{L}_{\mathcal{S}}^i \left( \boldsymbol{\theta}_{sh}, \boldsymbol{\theta}_{ns}^i \right) \|_2$ of task $i$, hence leading the model to the low-valued region for the loss of this task and its flatter region with a lower gradient norm magnitude.

Moreover, inspired from the derivation in Equation (6), we decompose the gradient $\boldsymbol{g}_{sh}^{i,\text{SAM}} = \boldsymbol{g}_{sh}^{i,\text{loss}} + \boldsymbol{g}_{sh}^{i,\text{flat}}$ where we define $\boldsymbol{g}_{sh}^{i,\text{loss}} := \nabla_{\boldsymbol{\theta}_{sh}} \mathcal{L}_{\mathcal{S}}^i \left( \boldsymbol{\theta}_{sh}, \boldsymbol{\theta}_{ns}^i \right)$ and $\boldsymbol{g}_{sh}^{i,\text{flat}} := \boldsymbol{g}_{sh}^{i,\text{SAM}} - \boldsymbol{g}_{sh}^{i,\text{loss}}$. As aforementioned, the purpose of the negative gradient $-\boldsymbol{g}_{sh}^{i,\text{loss}}$ is to orient the model to minimize the loss of the task $i$, while $-\boldsymbol{g}_{sh}^{i,\text{flat}}$ navigates the model to the task $i$'s flatter region.

Therefore, the SAM gradients $\boldsymbol{g}_{sh}^{i,\text{SAM}}, i \in [m]$ constitute two components with different purposes. To mitigate the possible confliction and interference of the two components when aggregating, we propose to aggregate the low-loss components solely and then the flat components solely. Specifically, to find a common direction that leads the joint low-valued losses for all tasks and the joint flatter region for them, we first combine the gradients $\boldsymbol{g}_{sh}^{i,\text{loss}}, i \in [m]$ and the gradients $\boldsymbol{g}_{sh}^{i,\text{flat}}, i \in [m]$, then

add the two aggregated gradients, and finally update the shared part as:

$$\boldsymbol{g}_{sh}^{\text{loss}} = \text{gradient\_aggregate}(\boldsymbol{g}_{sh}^{1,\text{loss}}, \ldots, \boldsymbol{g}_{sh}^{m,\text{loss}}),$$

$$\boldsymbol{g}_{sh}^{\text{flat}} = \text{gradient\_aggregate}(\boldsymbol{g}_{sh}^{1,\text{flat}}, \ldots, \boldsymbol{g}_{sh}^{m,\text{flat}}),$$

$$\boldsymbol{g}_{sh}^{\text{SAM}} = \boldsymbol{g}_{sh}^{\text{loss}} + \boldsymbol{g}_{sh}^{\text{flat}}; \; \boldsymbol{\theta}_{sh} = \boldsymbol{\theta}_{sh} - \eta \boldsymbol{g}_{sh}^{\text{SAM}},$$

Finally, the key steps of our proposed framework are summarized in Algorithm 1 and the overall schema of our proposed method is demonstrated in Figure 1.

---

**Algorithm 1** Sharpness minimization for multi-task learning

---

**Input:** Model parameter $\boldsymbol{\theta} = [\boldsymbol{\theta}_{sh}, \boldsymbol{\theta}_{ns}^{1:m}]$, perturbation radius $\rho = [\rho_{sh}, \rho_{ns}]$, step size $\eta$ and a list of $m$ differentiable loss functions $\left\{\mathcal{L}^i\right\}_{i=1}^m$.

**Output:** Updated parameter $\boldsymbol{\theta}^*$

1: **for** task $i \in [m]$ **do**
2:      Compute gradient $\boldsymbol{g}_{sh}^{i,\text{loss}}, \boldsymbol{g}_{ns}^i \leftarrow \nabla_{\boldsymbol{\theta}} \mathcal{L}^i(\boldsymbol{\theta})$
3:      Worst-case perturbation direction

$$\boldsymbol{\epsilon}_{sh}^i = \rho_{sh} \cdot \boldsymbol{g}_{sh}^{i,\text{loss}} / \left\|\boldsymbol{g}_{sh}^{i,\text{loss}}\right\| \text{ and } \boldsymbol{\epsilon}_{ns}^i = \rho_{ns} \cdot \boldsymbol{g}_{ns}^i / \left\|\boldsymbol{g}_{ns}^i\right\|$$

4:      Approximate SAM's gradient

$$\boldsymbol{g}_{sh}^{i,\text{SAM}} = \nabla_{\boldsymbol{\theta}_{sh}} \mathcal{L}^i(\boldsymbol{\theta}_{sh} + \boldsymbol{\epsilon}_{sh}^i, \boldsymbol{\theta}_{ns}^i) \text{ and } \boldsymbol{g}_{ns}^{i,\text{SAM}} = \nabla_{\boldsymbol{\theta}_{ns}^i} \mathcal{L}^i(\boldsymbol{\theta}_{sh}, \boldsymbol{\theta}_{ns}^i + \boldsymbol{\epsilon}_{ns}^i)$$

5:      Compute flat gradient

$$\boldsymbol{g}_{sh}^{i,\text{flat}} = \boldsymbol{g}_{sh}^{i,\text{SAM}} - \boldsymbol{g}_{sh}^{i,\text{loss}}$$

6: **end for**
7: Calculate combined update gradients:

$$\boldsymbol{g}_{sh}^{\text{loss}} = \text{gradient\_aggregate}(\boldsymbol{g}_{sh}^{1,\text{loss}}, \boldsymbol{g}_{sh}^{2,\text{loss}}, \ldots, \boldsymbol{g}_{sh}^{m,\text{loss}})$$

$$\boldsymbol{g}_{sh}^{\text{flat}} = \text{gradient\_aggregate}(\boldsymbol{g}_{sh}^{1,\text{flat}}, \boldsymbol{g}_{sh}^{2,\text{flat}}, \ldots, \boldsymbol{g}_{sh}^{m,\text{flat}})$$

8: Calculate shared gradient update $\boldsymbol{g}_{sh}^{\text{SAM}} = \boldsymbol{g}_{sh}^{\text{loss}} + \boldsymbol{g}_{sh}^{\text{flat}}$
9: Update model parameter

$$\boldsymbol{\theta}^* = [\boldsymbol{\theta}_{sh}, \boldsymbol{\theta}_{ns}^{1:m}] - \eta[\boldsymbol{g}_{sh}^{\text{SAM}}, \boldsymbol{g}_{ns}^{1:m,\text{SAM}}]$$

---

## 5 EXPERIMENTS

In this section, we compare our proposed method against other state-of-the-art methods of multi-task learning in different scenarios, ranging from image classification to scene understanding problems. Refer to the appendix for the detailed settings used for each dataset and additional experiments.

**Datasets and Baselines.** Our proposed method is evaluated on four MTL benchmarks including Multi-MNIST (Lin et al., 2019), CelebA (Liu et al., 2015) for visual classification, and NYUv2 (Silberman et al., 2012), CityScapes (Cordts et al., 2016) for scene understanding. We show how our framework can boost the performance of gradient-based MTL methods by comparing *vanilla* MGDA (Sener & Koltun, 2018), PCGrad (Yu et al., 2020), CAGrad (Liu et al., 2021a) and IMTL (Liu et al., 2020) to their flat-based versions F-MGDA, F-PCGrad, F-CAGrad and F-IMTL. We also add single task learning (STL) baseline for each dataset.

### 5.1 IMAGE CLASSIFICATION

**Multi-MNIST.** Following the protocol of Sener & Koltun (2018), we set up three Multi-MNIST experiments with the ResNet18 (He et al., 2016) backbone, namely: MultiFashion, MultiMNIST and MultiFashion+MNIST. In each dataset, two images are sampled uniformly from the MNIST (LeCun et al., 1998) or Fashion-MNIST (Xiao et al., 2017), then one is placed on the top left and the other is on the bottom right. We thus obtain a two-task learning that requires predicting the categories of the digits or fashion items on the top left (task 1) and on the bottom right (task 2) respectively.

As summarized in Table 1, we can see that seeking flatter regions for all tasks can improve the performance of all the baselines across all three datasets. Especially, flat-based methods achieve the highest score for each task and for the average, outperforming STL by $1.2\%$ on MultiFashion and MultiMNIST. We conjecture that the discrepancy between gradient update trajectories to classify digits from MNIST and fashion items from FashionMNIST has resulted in the fruitless performance of baselines, compared to STL on MultiFashion+MNIST. Even if there exists dissimilarity between tasks, our best obtained average accuracy when applying our method to CAGrad is just slightly lower than STL ($< 0.4\%$) while employing a single model only.

Table 1: Evaluation of different methods on three Multi-MNIST datasets. Rows with flat-based minimizers are shaded. Bold numbers denote higher accuracy between flat-based methods and their baselines. * denotes the highest accuracy (except for STL, since it unfairly exploits multiple neural networks). We also use arrows to indicate that the higher is the better (↑) or vice-versa (↓).

| Method | MultiFashion | | | MultiMNIST | | | MultiFashion+MNIST | | |
|---|---|---|---|---|---|---|---|---|---|
| | Task 1 ↑ | Task 2 ↑ | Average ↑ | Task 1 ↑ | Task 2 ↑ | Average ↑ | Task 1 ↑ | Task 2 | Average ↑ |
| STL | $87.10 \pm 0.09$ | $86.20 \pm 0.06$ | $86.65 \pm 0.02$ | $95.33 \pm 0.08$ | $94.16 \pm 0.04$ | $94.74 \pm 0.06$ | $98.40 \pm 0.02$ | $89.42 \pm 0.03$ | $93.91 \pm 0.02$ |
| MGDA | $86.76 \pm 0.09$ | $85.78 \pm 0.36$ | $86.27 \pm 0.22$ | $95.62 \pm 0.02$ | $94.49 \pm 0.10$ | $95.05 \pm 0.06$ | $97.24 \pm 0.04$ | $88.19 \pm 0.13$ | $92.72 \pm 0.07$ |
| F-MGDA | $\mathbf{88.12 \pm 0.11}$ | $\mathbf{87.35 \pm 0.11}$ | $\mathbf{87.73 \pm 0.09}$ | $\mathbf{96.37 \pm 0.06}$ | $\mathbf{94.99 \pm 0.06}$ | $\mathbf{95.68 \pm 0.00}$ | $\mathbf{97.30 \pm 0.09}$ | $\mathbf{89.26 \pm 0.14}$ | $\mathbf{93.28 \pm 0.03}$ |
| PCGrad | $86.93 \pm 0.17$ | $86.20 \pm 0.14$ | $86.57 \pm 0.12$ | $95.71 \pm 0.03$ | $94.41 \pm 0.02$ | $95.06 \pm 0.02$ | $97.12 \pm 0.16$ | $88.45 \pm 0.08$ | $92.78 \pm 0.11$ |
| F-PCGrad | $\mathbf{88.17 \pm 0.14}$ | $\mathbf{87.35 \pm 0.27}$ | $\mathbf{87.76 \pm 0.07}$ | $\mathbf{96.49 \pm 0.06}$ | $\mathbf{95.34 \pm 0.10}$ | $\mathbf{95.92 \pm 0.07}$ | $\mathbf{97.65 \pm 0.06}$ | $\mathbf{89.35 \pm 0.07^*}$ | $\mathbf{93.50 \pm 0.01}$ |
| CAGrad | $86.99 \pm 0.17$ | $86.04 \pm 0.15$ | $86.51 \pm 0.16$ | $95.62 \pm 0.05$ | $94.39 \pm 0.04$ | $95.01 \pm 0.04$ | $97.19 \pm 0.06$ | $88.18 \pm 0.14$ | $92.68 \pm 0.04$ |
| F-CAGrad | $\mathbf{88.19 \pm 0.19^*}$ | $\mathbf{87.45 \pm 0.13}$ | $\mathbf{87.82 \pm 0.10^*}$ | $\mathbf{96.54 \pm 0.02}$ | $\mathbf{95.36 \pm 0.04^*}$ | $\mathbf{95.95 \pm 0.01^*}$ | $\mathbf{97.82 \pm 0.05^*}$ | $\mathbf{89.26 \pm 0.22}$ | $\mathbf{93.54 \pm 0.13^*}$ |
| IMTL | $87.35 \pm 0.22$ | $86.45 \pm 0.09$ | $86.90 \pm 0.15$ | $95.93 \pm 0.09$ | $94.63 \pm 0.13$ | $95.28 \pm 0.02$ | $97.47 \pm 0.06$ | $88.46 \pm 0.11$ | $92.97 \pm 0.03$ |
| F-IMTL | $\mathbf{88.1 \pm 0.10}$ | $\mathbf{87.5 \pm 0.04^*}$ | $\mathbf{87.80 \pm 0.06}$ | $\mathbf{96.55 \pm 0.07^*}$ | $\mathbf{95.16 \pm 0.05}$ | $\mathbf{95.85 \pm 0.05}$ | $\mathbf{97.59 \pm 0.12}$ | $\mathbf{88.99 \pm 0.08}$ | $\mathbf{93.29 \pm 0.02}$ |

Interestingly, our proposed MTL training method also helps improve model calibration performance by mitigating the over-confident phenomenon of deep neural networks. As can be seen from Figure 2, our method produces high-entropy predictions that represent its uncertainty, ERM-based method outputs high confident predictions on both in and out-of-domain data. More details about model calibration improvement can be found in the appendix.

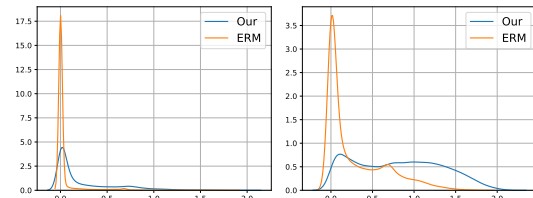

Figure 2: Entropy distributions of ResNet18 on in-domain set (left) and out-of-domain set (right).

**CelebA.** CelebA (Liu et al., 2018) is a face dataset, which consists of 200K celebrity facial photos with 40 attributes. Similar to (Sener & Koltun, 2018), each attribute forms a binary classification problem, thus a 40-class multi-label classification problem is constructed.

Table 2 shows the average errors over 40 tasks of the methods with linear scalarization (LS) and Uncertainty weighting (UW) (Kendall et al., 2018) being included to serve as comparative baselines. The best results in each pair and among all are highlighted using **bold** font and *, respectively. When the number of tasks is large, flat region seeking still consistently shows its advantages and the lowest average accuracy error is achieved by F-CAGrad. Interestingly, when the optimizer is aware of flat minima, the gaps between PCGrad, IMTL and CAGrad, (8.23, 8.24 vs 8.22), are smaller than those using conventional ERM training, (8.69, 8.88 and 8.52). This might be due to the better aggregation of tasks' gradients, which means that the conflict between these gradients is likely to be reduced when the shared parameters approach the common flat region of all tasks.

Table 2: Mean of error per category of MTL algorithms in multi-label classification on CelebA dataset.

| Method | Average error ↓ |
|---|---|
| STL | 8.77 |
| LS | 9.99 |
| UW | 9.66 |
| MGDA | 9.96 |
| F-MGDA | **9.22** |
| PCGrad | 8.69 |
| F-PCGrad | **8.23** |
| CAGrad | 8.52 |
| F-CAGrad | **8.22***|
| IMTL | 8.88 |
| F-IMTL | **8.24** |

## 5.2 Scene Understanding

Two datasets used in this sub-section are NYUv2 (Silberman et al., 2012) and CityScapes (Cordts et al., 2016). NYUv2 is an indoor scene dataset that contains 3 tasks: 13-class semantic segmentation, depth estimation, and surface normal prediction. In CityScapes, there are 19 classes of street-view images, which are coarsened into 7 categories to create two tasks: semantic segmentation and depth estimation. For these two experiments, we additionally include several recent MTL methods, namely, scale-invariant (SI), random loss weighting (RLW), Dynamic Weight Average (DWA) (Liu et al., 2019a), GradDrop (Chen et al., 2020), and Nash-MTL (Navon et al., 2022) whose results are taken from (Navon et al., 2022). Details of each baseline can be found in the appendix. Also following the standard protocol used in (Liu et al., 2019a; 2021a; Navon et al., 2022), Multi-Task Attention Network (Liu et al., 2019a) is employed on top of the SegNet architecture (Badrinarayanan et al., 2017), our presented results are averaged over the last 10 epochs to align with previous work.

**Evaluation metric.** In this experiment, we have to deal with different task types rather than one only as in the case of image classification. Since each of them has its own set of metrics. We thus mark the overall performance of comparative methods by reporting their relative task improvement (Maninis et al., 2019) throughout this section. Let $M_i$ and $S_i$ be the metrics obtained by the main and the

single-task learning (STL) model, respectively, the relative task improvement on $i$-th task is mathematically given by: $\Delta_i := 100 \cdot (-1)^{l_i} (M_i - S_i)/S_i$, where $l_i = 1$ if a lower value for the $i$-th criterion is better and 0 otherwise. We depict our results by the average relative task improvement $\Delta m\% = \frac{1}{m}\sum_{i=1}^{m}\Delta_i$.

**CityScapes.** In Table 3, the positive effect of seeking flat regions is consistently observed in all metrics and baselines. In particular, the relative improvements of MGDA and IMTL are significantly boosted, achieving the highest and second-best $\Delta m\%$ scores, respectively. The segmentation scores of PCGrad, CAGrad and IMTL even surpass STL. Intriguingly, MGDA biases to the depth estimation objective, leading to the predominant performance on that task, similar patterns appear in Liu et al., 2020 and the below NYUv2 experiment.

Table 3: Test performance for two-task CityScapes: semantic segmentation and depth estimation.* denotes the best score for each task's metrics.

| Method | Segmentation | | Depth | | |
| --- | --- | --- | --- | --- | --- |
| | mIoU ↑ | Pix Acc ↑ | Abs Err ↓ | Rel Err↓ | $\Delta m\%$ ↓ |
| STL | 74.01 | 93.16 | 0.0125 | 27.77 | |
| LS | 75.18 | 93.49 | 0.0155 | 46.77 | 22.60 |
| SI | 70.95 | 91.73 | 0.0161 | 33.83 | 14.11 |
| RLW | 74.57 | 93.41 | 0.0158 | 47.79 | 24.38 |
| DWA | 75.24 | 93.52 | 0.0160 | 44.37 | 21.45 |
| UW | 72.02 | 92.85 | 0.0140 | 30.13* | 5.89 |
| GradDrop | 75.27 | 93.53 | 0.0157 | 47.54 | 23.73 |
| Nash-MTL | 75.41 | 93.66 | 0.0129 | 35.02 | 6.82 |
| MGDA | 68.84 | 91.54 | 0.0309 | 33.50 | 44.14 |
| F-MGDA | **73.77** | **93.12** | **0.0129** | **27.44*** | **0.67*** |
| PCGrad | 75.13 | 93.48 | 0.0154 | 42.07 | 18.29 |
| F-PCGrad | **75.77** | **93.67** | **0.0144** | **39.60** | **13.65** |
| CAGrad | 75.16 | 93.48 | 0.0141 | 37.60 | 11.64 |
| F-CAGrad | **76.02** | **93.72** | **0.0134** | **34.64** | **7.25** |
| IMTL | 75.33 | 93.49 | 0.0135 | 38.41 | 11.10 |
| F-IMTL | **76.63*** | **93.76*** | **0.0124*** | **31.17** | **1.87** |

**NYUv2.** Table 4 shows each task's results and the relative improvements over STL of different methods. Generally, the flat-based versions obtain comparable or higher results on most of the metrics, except for MGDA at the segmentation task, in which F-MGDA notably decreases the $mIoU$ score. However, it does significantly help other tasks, which contributes to the overall MGDA's relative improvement, from $1.38\%$ being worse than STL to $0.33\%$ being higher. Remarkably, F-CAGrad and F-IMTL outperform their competitors by large margins across all tasks, resulting in the top two relative improvements, $3.78\%$ and $4.77\%$.

Table 4: Test performance for three-task NYUv2 of Segnet (Badrinarayanan et al., 2017): semantic segmentation, depth estimation, and surface normal. Using the proposed procedure in conjunction with gradient-based multi-task learning methods consistently advances their overall performance.

| | Segmentation | | Depth | | Surface Normal | | | | | $\Delta m\%$ ↓ |
| --- | --- | --- | --- | --- | --- | --- | --- | --- | --- | --- |
| | | | | | Angle Distance ↓ | | Within $t°$ ↑ | | | |
| | mIoU ↑ | Pix Acc ↑ | Abs Err ↓ | Rel Err ↓ | Mean | Median | 11.25 | 22.5 | 30 | |
| STL | 38.30 | 63.76 | 0.6754 | 0.2780 | 25.01 | 19.21 | 30.14 | 57.20 | 69.15 | 0.00 |
| LS | 39.29 | 65.33 | 0.5493 | 0.2263 | 28.15 | 23.96 | 22.09 | 47.50 | 61.08 | 5.59 |
| SI | 38.45 | 64.27 | 0.5354 | 0.2201 | 27.60 | 23.37 | 22.53 | 48.57 | 62.32 | 4.39 |
| RLW | 37.17 | 63.77 | 0.5759 | 0.2410 | 28.27 | 24.18 | 22.26 | 47.05 | 60.62 | 7.78 |
| DWA | 39.11 | 65.31 | 0.5510 | 0.2285 | 27.61 | 23.18 | 24.17 | 50.18 | 62.39 | 3.57 |
| UW | 36.87 | 63.17 | 0.5446 | 0.2260 | 27.04 | 22.61 | 23.54 | 49.05 | 63.65 | 4.05 |
| GradDrop | 39.39 | 65.12 | 0.5455 | 0.2279 | 27.48 | 22.96 | 23.38 | 49.44 | 62.87 | 3.58 |
| Nash-MTL | 40.13 | 65.93 | 0.5261* | 0.2171 | 25.26 | 20.08 | 28.4 | 55.47 | 68.15 | −4.04 |
| MGDA | **30.47** | **59.90** | **0.6070** | 0.2555 | 24.88 | 19.45 | 29.18 | 56.88 | 69.36 | 1.38 |
| F-MGDA | 26.42 | 58.78 | 0.6078 | **0.2353** | **24.34*** | **18.45*** | **31.64*** | **58.86*** | **70.50*** | **−0.33** |
| PCGrad | 38.06 | 64.64 | 0.5550 | 0.2325 | 27.41 | **22.80** | **23.86** | **49.83** | **63.14** | 3.97 |
| F-PCGrad | **40.05** | **65.42** | **0.5429** | **0.2243** | **27.38** | 23.00 | 23.47 | 49.35 | 62.74 | **3.14** |
| CAGrad | 39.79 | 65.49 | 0.5486 | 0.2250 | 26.31 | 21.58 | 25.61 | 52.36 | 65.58 | 0.20 |
| F-CAGrad | **40.93*** | **66.68*** | **0.5285** | **0.2162** | **25.43** | **20.39** | **27.99** | **54.82** | **67.56** | **−3.78** |
| IMTL | 39.35 | 65.60 | 0.5426 | 0.2256 | 26.02 | 21.19 | 26.2 | 53.13 | 66.24 | −0.76 |
| F-IMTL | **40.42** | **65.61** | **0.5389** | **0.2121*** | **25.03** | **19.75** | **28.90** | **56.19** | **68.72** | **−4.77*** |

## 5.3 ABLATION STUDY

So far, our proposed technique has shown state-of-the-art performances under different settings, we now investigate in more detailed how it affects conventional training by inspecting loss surfaces and model robustness. Similar patterns are observed in other experiments and given in the appendix.

**Task conflict.** To empirically confirm that tasks' gradients are less conflicted when the model is driven to the flat regions, we measure the gradient conflict and present the result in Figure 3a. While the percentage of gradient conflict of ERM increases to more than $50\%$, ours decreases and approaches $0\%$. This reduction in gradient conflict is also the goal of recent gradient-based MTL methods in mitigating negative transfer between tasks (Yu et al., 2020; Zhu et al., 2022; Wang et al., 2020).

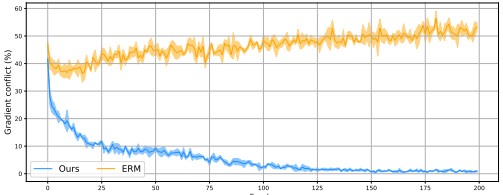
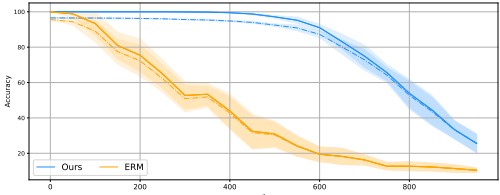

(a) Proportion of conflict between per-task gradients ($\boldsymbol{g}^{1,\text{loss}} \cdot \boldsymbol{g}^{2,\text{loss}} < 0$) on MultiFashion+MNIST dataset.

(b) Accuracy within $r-$radius ball. Solid/dashed lines denote performance on train/test sets, respectively.

**Model robustness against noise.** To verify that SAM can orient the model to the common flat and low-loss region of all tasks, we measure the model performance within a $r$-radius Euclidean ball. To be more specific, we perturb parameters of two converged models by $\epsilon$, which lies in a $r$-radius ball and plot the accuracy of the perturbed models of each task as we increase $r$ from $0$ to $1000$. At each value of $r$, $10$ different models around the $r-$radius ball of the converged model are sampled.

In Figure 3b, the accuracy of the model trained using our method remains at a high level when noise keeps increasing until $r = 800$. This also gives evidence that our model found a region that changes slowly in loss. By contrast, the naively trained model loses its predictive capabilities as soon as the noise appears and becomes a dummy classifier that attains $10\%$ accuracy in a 10-way classification.

**Aggregation strategies comparison.** Table 5 provides a comparison between the **direct** aggregation on $\{\boldsymbol{g}_{sh}^{i,\text{SAM}}\}_{i=1}^{m}$ and **individual** aggregation on $\{\boldsymbol{g}_{sh}^{i,\text{flat}}\}_{i=1}^{m}$ and $\{\boldsymbol{g}_{sh}^{i,\text{loss}}\}_{i=1}^{m}$ (our method).

Compared to the naive approach, in which per-task SAM gradients are directly aggregated, our decomposition approach consistently improves performance by a large margin across all tasks. This result reinforces the rationale behind separately aggregating low-loss directions and flat directions.

Table 5: Two aggregation strategies on CityScapes.

| | Segmentation | | Depth | | |
|---|---|---|---|---|---|
| Method | mIoU ↑ | Pix Acc ↑ | Abs Err ↓ | Rel Err ↓ | $\boldsymbol{\Delta m}\% \downarrow$ |
| ERM | 68.84 | 91.54 | 0.0309 | 33.50 | 44.14 |
| Ours (**direct**) | 68.93 | 91.41 | 0.0130 | 31.37 | 6.43 |
| Ours (**individual**) | **73.77** | **93.12** | **0.0129** | **27.44**$^*$ | **0.67** |

**Visualization of the loss landscapes.** Following Li et al. (2018), we plot the loss surfaces at convergence after training Resnet18 from scratch on the MultiMNIST dataset. Test loss surfaces of checkpoints that have the highest validation accuracy scores are shown in Figure 4.

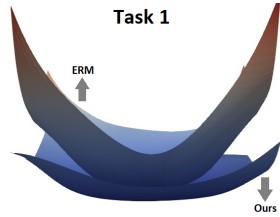
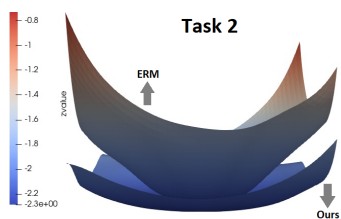

Figure 4: Visualization of test loss surfaces with standard ERM training and when applying our method. The coordinate plane axes are two random sampled orthogonal Gaussian perturbations.

We can clearly see that the solution found by our proposed method not only mitigates the test loss sharpness for both tasks but also can intentionally reduce the test loss value itself, in comparison to traditional ERM. This is a common behavior when using flat minimizers as the gap between train and test performance has been narrowed (Izmailov et al., 2018; Kaddour et al., 2022).

## 6 CONCLUSION

In this work, we have presented a general framework that can be incorporated into current multi-task learning methods following the gradient balancing mechanism. The core ideas of our proposed method are the employment of flat minimizers in the context of MTL and proving that they can help enhance previous works both theoretically and empirically. Concretely, our method goes beyond optimizing per-task objectives solely to yield models that have both low errors and high generalization capabilities. On the experimental side, the efficacy of our method is demonstrated on a wide range of commonly used MTL benchmarks, in which ours consistently outperforms comparative methods.

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

## A  APPENDIX

Due to space constraints, some details were omitted from the main paper. We therefore include additional theoretical developments (section B) and experimental results (section D) in this appendix.

## B  OUR THEORY DEVELOPMENT

This section contains the proofs and derivations of our theory development to support the main submission.

We first start with the following theorem, which is inspired by the general PAC-Bayes in (Alquier et al., 2016).

**Theorem 2.** *With the assumption that adding Gaussian perturbation will raise the test error:* $\mathcal{L}_{\mathcal{D}}(\boldsymbol{\theta}) \leq \mathbb{E}_{\epsilon \sim \mathcal{N}(0, \sigma^2 \mathbb{I})} [\mathcal{L}_{\mathcal{D}}(\boldsymbol{\theta} + \boldsymbol{\epsilon})]$. *Let* $T$ *be the number of parameter* $\boldsymbol{\theta}$, *and* $N$ *be the cardinality of* $\mathcal{S}$, *then the following inequality is true with the probability* $1 - \delta$:

$$\mathcal{L}_{\mathcal{D}}(\boldsymbol{\theta}) \leq \mathbb{E}_{\epsilon \sim \mathcal{N}(0, \sigma^2 \mathbb{I})} [\mathcal{L}_{\mathcal{S}}(\boldsymbol{\theta} + \boldsymbol{\epsilon})] + \frac{1}{\sqrt{N}} \left[ \frac{1}{2} + \frac{T}{2} \log \left( 1 + \frac{||\boldsymbol{\theta}||^2}{T\sigma^2} \right) + \log \frac{1}{\delta} + 6 \log(N + T) + \frac{L^2}{8} \right]$$

*where* $L$ *is the upper-bound of the loss function.*

*Proof.* We use the PAC-Bayes theory for $P = \mathcal{N}(\mathbf{0}, \sigma_P^2 \mathbb{I}_T)$ and $Q = \mathcal{N}(\boldsymbol{\theta}, \sigma^2 \mathbb{I}_T)$ are the prior and posterior distributions, respectively.

By using the bound in (Alquier et al., 2016), with probability at least $1 - \delta$ and for all $\beta > 0$, we have:

$$\mathbb{E}_{\boldsymbol{\theta} \sim Q} [\mathcal{L}_{\mathcal{D}}(\boldsymbol{\theta})] \leq \mathbb{E}_{\boldsymbol{\theta} \sim Q} [\mathcal{L}_{\mathcal{S}}(\boldsymbol{\theta})] + \frac{1}{\beta} \left[ \mathsf{KL}(Q \| P) + \log \frac{1}{\delta} + \Psi(\beta, N) \right],$$

where we have defined:

$$\Psi(\beta, N) = \log \mathbb{E}_P \mathbb{E}_{\mathcal{S}} \left[ \exp \left\{ \beta \big( \mathcal{L}_{\boldsymbol{D}}(\boldsymbol{\theta}) - \mathcal{L}_{\mathcal{S}}(\boldsymbol{\theta}) \big) \right\} \right]$$

Note that the loss function is bounded by $L$, according to Hoeffding's lemma, we have:

$$\Psi(\beta, N) \leq \frac{\beta^2 L^2}{8N}.$$

By Cauchy inequality:

$$\frac{1}{\sqrt{N}} \left[ \frac{T}{2} \log \left( 1 + \frac{||\boldsymbol{\theta}||^2}{T\sigma^2} \right) + \frac{L^2}{8} \right] \geq \frac{L}{2\sqrt{N}} \sqrt{T \log \left( 1 + \frac{||\boldsymbol{\theta}||^2}{T\sigma^2} \right)} \geq L,$$

which means that the theorem is proved since the loss function is upper bounded by $L$, following assumptions.

Now, we only need to prove the theorem under the case: $||\boldsymbol{\theta}||^2 \leq T\sigma^2 \left[ \exp \frac{4N}{T} - 1 \right]$.

We need to specify $P$ in advance since it is a prior distribution. However, we do not know in advance the value of $\boldsymbol{\theta}$ that affects the KL divergence term. Hence, we build a family of distribution $P$ as follows:

$$\mathfrak{P} = \left\{ P_j = \mathcal{N}(\mathbf{0}, \sigma_{P_j}^2 \mathbb{I}_T) : \sigma_{P_j}^2 = c \exp \left( \frac{1-j}{T} \right), c = \sigma^2 \left( 1 + \exp \frac{4N}{T} \right), j = 1, 2, \ldots \right\}.$$

Set $\delta_j = \frac{6\delta}{\pi^2 j^2}$, the below inequality holds with probability at least $1 - \delta_j$:

$$\mathbb{E}_{\boldsymbol{\theta} \sim Q} [\mathcal{L}_{\mathcal{D}}(\boldsymbol{\theta})] \leq \mathbb{E}_{\boldsymbol{\theta} \sim Q} [\mathcal{L}_{\mathcal{S}}(\boldsymbol{\theta})] + \frac{1}{\beta} \left[ \mathsf{KL}(Q \| P_j) + \log \frac{1}{\delta_j} + \frac{\beta^2 L^2}{8N} \right].$$

Or it can be written as:

$$\mathbb{E}_{\epsilon \sim \mathcal{N}(0,\sigma^2\mathbb{I})}\left[\mathcal{L}_\mathcal{D}(\boldsymbol{\theta} + \boldsymbol{\epsilon})\right] \leq \mathbb{E}_{\epsilon \sim \mathcal{N}(0,\sigma^2\mathbb{I})}\left[\mathcal{L}_\mathcal{S}(\boldsymbol{\theta} + \boldsymbol{\epsilon})\right] + \frac{1}{\beta}\left[\mathsf{KL}(Q\|P_j) + \log\frac{1}{\delta_j} + \frac{\beta^2 L^2}{8N}\right].$$

Thus, with probability $1 - \delta$ the above inequalities hold for all $P_j$. We choose:

$$j^* = \left\lfloor 1 + T\log\left(\frac{\sigma^2\left(1 + \exp\{4N/T\}\right)}{\sigma^2 + \|\boldsymbol{\theta}\|^2/T}\right)\right\rfloor.$$

Since $\frac{\|\boldsymbol{\theta}\|^2}{T} \leq \sigma^2\left[\exp\frac{4N}{T} - 1\right]$, we get $\sigma^2 + \frac{\|\boldsymbol{\theta}\|^2}{T} \leq \sigma^2\exp\frac{4N}{T}$, thus $j^*$ is well-defined. We also have:

$$T\log\frac{c}{\sigma^2 + \|\boldsymbol{\theta}\|^2/T} \qquad \leq j^* \qquad \leq 1 + T\log\frac{c}{\sigma^2 + \|\boldsymbol{\theta}\|^2/T}$$

$$\Rightarrow \quad \log\frac{c}{\sigma^2 + \|\boldsymbol{\theta}\|^2/T} \qquad \leq \frac{j^*}{T} \qquad \leq \frac{1}{T} + \log\frac{c}{\sigma^2 + \|\boldsymbol{\theta}\|^2/T}$$

$$\Rightarrow \quad -\frac{1}{T} + \log\frac{\sigma^2 + \|\boldsymbol{\theta}\|^2/T}{c} \leq \frac{-j^*}{T} \qquad \leq \log\frac{\sigma^2 + \|\boldsymbol{\theta}\|^2/T}{c}$$

$$\Rightarrow \quad e^{-1/T}\frac{\sigma^2 + \|\boldsymbol{\theta}\|^2/T}{c} \qquad \leq e^{-j^*/T} \leq \frac{\sigma^2 + \|\boldsymbol{\theta}\|^2/T}{c}$$

$$\Rightarrow \quad \sigma^2 + \frac{\|\boldsymbol{\theta}\|^2}{T} \qquad \leq ce^{\frac{1-j^*}{T}} \leq e^{\frac{1}{T}}\left(\sigma^2 + \frac{\|\boldsymbol{\theta}\|^2}{T}\right)$$

$$\Rightarrow \quad \sigma^2 + \frac{\|\boldsymbol{\theta}\|^2}{T} \qquad \leq \sigma^2_{P_{j^*}} \leq e^{\frac{1}{T}}\left(\sigma^2 + \frac{\|\boldsymbol{\theta}\|^2}{T}\right).$$

Hence, we have:

$$\mathsf{KL}(Q\|P_{j^*}) = \frac{1}{2}\left[\frac{T\sigma^2 + \|\boldsymbol{\theta}\|^2}{\sigma^2_{P_{j^*}}} - T + T\log\frac{\sigma^2_{P_{j^*}}}{\sigma^2}\right]$$

$$\leq \frac{1}{2}\left[\frac{T\sigma^2 + \|\boldsymbol{\theta}\|^2}{\sigma^2 + \|\boldsymbol{\theta}\|^2/T} - T + T\log\frac{e^{1/T}\left(\sigma^2 + \|\boldsymbol{\theta}\|^2/T\right)}{\sigma^2}\right]$$

$$\leq \frac{1}{2}\left[1 + T\log\left(1 + \frac{\|\boldsymbol{\theta}\|^2}{T\sigma^2}\right)\right].$$

For the term $\log\frac{1}{\delta_{j^*}}$, use the inequality $\log(1 + e^t) \leq 1 + t$ for $t > 0$:

$$\log\frac{1}{\delta_{j^*}} = \log\frac{(j^*)^2\pi^2}{6\delta} = \log\frac{1}{\delta} + \log\left(\frac{\pi^2}{6}\right) + 2\log(j^*)$$

$$\leq \log\frac{1}{\delta} + \log\frac{\pi^2}{6} + 2\log\left(1 + T\log\frac{\sigma^2\left(1 + \exp(4N/T)\right)}{\sigma^2 + \|\boldsymbol{\theta}\|^2/T}\right)$$

$$\leq \log\frac{1}{\delta} + \log\frac{\pi^2}{6} + 2\log\left(1 + T\log\left(1 + \exp(4N/T)\right)\right)$$

$$\leq \log\frac{1}{\delta} + \log\frac{\pi^2}{6} + 2\log\left(1 + T\left(1 + \frac{4N}{T}\right)\right)$$

$$\leq \log\frac{1}{\delta} + \log\frac{\pi^2}{6} + \log(1 + T + 4N).$$

Choosing $\beta = \sqrt{N}$, with probability at least $1 - \delta$ we get:

$$\frac{1}{\beta}\left[\mathsf{KL}(Q\|P_{j^*}) + \log\frac{1}{\delta_{j^*}} + \frac{\beta^2 L^2}{8N}\right]$$

$$\leq \frac{1}{\sqrt{N}}\left[\frac{1}{2} + \frac{T}{2}\log\left(1 + \frac{\|\boldsymbol{\theta}\|^2}{T\sigma^2}\right) + \log\frac{1}{\delta} + 6\log(N + T)\right] + \frac{L^2}{8\sqrt{N}}.$$

Thus the theorem is proved. $\qquad\qquad\square$

Back to our context of multi-task learning in which we have $m$ tasks with each task model: $\boldsymbol{\theta}^i = [\boldsymbol{\theta}_{sh}, \boldsymbol{\theta}_{ns}^i]$, we can prove the following theorem.

**Theorem 3.** *With the assumption that adding Gaussian perturbation will rise the test error:* $\mathcal{L}_{\mathcal{D}}(\boldsymbol{\theta}^i) \leq \mathbb{E}_{\epsilon \sim \mathcal{N}(0,\sigma^2\mathbb{I})}\left[\mathcal{L}_{\mathcal{D}}(\boldsymbol{\theta}^i + \boldsymbol{\epsilon})\right]$. *Let $T_i$ be the number of parameter $\boldsymbol{\theta}^i$ and $N$ be the cardinality of $\mathcal{S}$. We have the following inequality holds with probability $1 - \delta$ (over the choice of training set $\mathcal{S} \sim \mathcal{D}$):*

$$\left[\mathcal{L}_{\mathcal{D}}^i\left(\boldsymbol{\theta}^i\right)\right]_{i=1}^m \leq \left[\mathbb{E}_{\epsilon \sim \mathcal{N}(0,\sigma^2\mathbb{I})}\left[\mathcal{L}_{\mathcal{S}}(\boldsymbol{\theta}^i + \boldsymbol{\epsilon})\right] + f^i\left(\|\boldsymbol{\theta}^i\|_2^2\right)\right]_{i=1}^m, \tag{7}$$

*where*

$$f^i\left(\|\boldsymbol{\theta}^i\|_2^2\right) = \frac{1}{\sqrt{N}}\left[\frac{1}{2} + \frac{T_i}{2}\log\left(1 + \frac{\|\boldsymbol{\theta}\|^2}{T_i\sigma^2}\right) + \log\frac{1}{\delta} + 6\log(N + T_i) + \frac{L^2}{8}\right].$$

**Proof.** The result for the base case $m = 1$ can be achieved by using Theorem 2 where $\xi = \delta$ and $f^1$ is defined accordingly. We proceed by induction, suppose that Theorem 3 is true for all $i \in [n]$ with probability $1 - \delta/2$, which also means:

$$\left[\mathcal{L}_{\mathcal{D}}^i\left(\boldsymbol{\theta}^i\right)\right]_{i=1}^n \leq \left[\mathbb{E}_{\epsilon \sim \mathcal{N}(0,\sigma\mathbb{I})}\left[\mathcal{L}_{\mathcal{S}}(\boldsymbol{\theta}^i + \boldsymbol{\epsilon})\right] + f^i\left(\|\boldsymbol{\theta}^i\|_2^2\right)\right]_{i=1}^n.$$

Using Theorem 2 for $\boldsymbol{\theta}^{n+1}$ and $\xi = \delta/2$, with probability $1 - \delta/2$, we have:

$$\mathcal{L}_{\mathcal{D}}^{n+1}\left(\boldsymbol{\theta}^{n+1}\right) \leq \mathbb{E}_{\epsilon \sim \mathcal{N}(0,\sigma\mathbb{I})}\left[\mathcal{L}_{\mathcal{S}}(\boldsymbol{\theta}^{n+1} + \boldsymbol{\epsilon})\right] + f^{n+1}\left(\|\boldsymbol{\theta}^{n+1}\|_2^2\right).$$

Using the inclusion–exclusion principle, with probability at least $1 - \delta$, we reach the conclusion for $m = n + 1$.

We next prove the result in the main paper. Let us begin by formally restating the main theorem as follows:

**Theorem 4.** *For any perturbation radius $\rho_{sh}, \rho_{ns} > 0$, with probability $1 - \delta$ (over the choice of training set $\mathcal{S} \sim \mathcal{D}$) we obtain:*

$$\left[\mathcal{L}_{\mathcal{D}}^i\left(\boldsymbol{\theta}^i\right)\right]_{i=1}^m \leq \max_{\|\boldsymbol{\epsilon}_{sh}\|_2 \leq \rho_{sh}}\left[\max_{\|\boldsymbol{\epsilon}_{ns}^i\|_2 \leq \rho_{ns}} \mathcal{L}_{\mathcal{S}}^i\left(\boldsymbol{\theta}_{sh} + \boldsymbol{\epsilon}_{sh}, \boldsymbol{\theta}_{ns}^i + \boldsymbol{\epsilon}_{ns}^i\right) + f^i\left(\|\boldsymbol{\theta}^i\|_2^2\right)\right]_{i=1}^m, \tag{8}$$

*where $f^i\left(\|\boldsymbol{\theta}^i\|_2^2\right)$ is defined the same as in Theorem 3.*

**Proof.** Theorem 3 gives us

$$\begin{aligned}
\left[\mathcal{L}_{\mathcal{D}}^i\left(\boldsymbol{\theta}^i\right)\right]_{i=1}^m &\leq \left[\mathbb{E}_{\epsilon \sim N(0,\sigma^2\mathbb{I})}\left[\mathcal{L}_{\mathcal{S}}^i\left(\boldsymbol{\theta}^i + \boldsymbol{\epsilon}\right)\right] + f^i\left(\|\boldsymbol{\theta}^i\|_2\right)\right]_{i=1}^m \\
&= \left[\int \mathbb{E}_{\boldsymbol{\epsilon}_{ns}^i}\left[\mathcal{L}_{\mathcal{S}}^i\left(\boldsymbol{\theta}_{sh} + \boldsymbol{\epsilon}_{sh}, \boldsymbol{\theta}_{ns}^i + \boldsymbol{\epsilon}_{ns}^i\right)\right] p\left(\boldsymbol{\epsilon}_{sh}\right) d\boldsymbol{\epsilon}_{sh} + f^i\left(\|\boldsymbol{\theta}^i\|_2\right)\right]_{i=1}^m \\
&= \mathbb{E}_{\boldsymbol{\epsilon}_{sh}}\left[\mathbb{E}_{\boldsymbol{\epsilon}_{ns}^i}\left[\mathcal{L}_{\mathcal{S}}^i\left(\boldsymbol{\theta}_{sh} + \boldsymbol{\epsilon}_{sh}, \boldsymbol{\theta}_{ns}^i + \boldsymbol{\epsilon}_{ns}^i\right)\right] + f^i\left(\|\boldsymbol{\theta}^i\|_2\right)\right]_{i=1}^m,
\end{aligned}$$

where $p(\boldsymbol{\epsilon}_{sh})$ is the density function of Gaussian distribution; $\boldsymbol{\epsilon}_{sh}$ and $\boldsymbol{\epsilon}_{ns}^i$ are drawn from their corresponding Gaussian distributions.

We have $\boldsymbol{\epsilon}_{ns}^i \sim N(0, \sigma^2\mathbb{I}_{ns})$ with the dimension $T_{i,ns}$, therefore $\|\boldsymbol{\epsilon}_{ns}^i\|$ follows the Chi-square distribution. As proven in (Laurent & Massart, 2000), we have for all $i$:

$$P\left(\|\boldsymbol{\epsilon}_{ns}^i\|_2^2 \geq T_{i,ns}\sigma^2 + 2\sigma^2\sqrt{T_{i,ns}t} + 2t\sigma^2\right) \leq e^{-t}, \forall t > 0$$

$$P\left(\|\boldsymbol{\epsilon}_{ns}^i\|_2^2 < T_{i,ns}\sigma^2 + 2\sigma^2\sqrt{T_{i,ns}t} + 2t\sigma^2\right) > 1 - e^{-t}$$

for all $t > 0$.

Select $t = \ln(\sqrt{N})$, we derive the following bound for the noise magnitude in terms of the perturbation radius $\rho_{ns}$ for all $i$:

$$P\left(\|\boldsymbol{\epsilon}_{ns}^i\|_2^2 \leq \sigma^2(2\ln(\sqrt{N}) + T_{i,ns} + 2\sqrt{T_{i,ns}\ln(\sqrt{N})})\right) > 1 - \frac{1}{\sqrt{N}}. \tag{9}$$

Moreover, we have $\epsilon_{sh} \sim N(0, \sigma^2 \mathbb{I}_{sh})$ with the dimension $T_{sh}$, therefore $\|\epsilon_{sh}\|$ follows the Chi-square distribution. As proven in (Laurent & Massart, 2000), we have:

$$P\left(\|\epsilon_{sh}\|_2^2 \geq T_{sh}\sigma^2 + 2\sigma^2\sqrt{T_{sh}t} + 2t\sigma^2\right) \leq e^{-t}, \forall t > 0$$

$$P\left(\|\epsilon_{sh}\|_2^2 < T_{sh}\sigma^2 + 2\sigma^2\sqrt{T_{sh}t} + 2t\sigma^2\right) > 1 - e^{-t}$$

for all $t > 0$.

Select $t = \ln(\sqrt{N})$, we derive the following bound for the noise magnitude in terms of the perturbation radius $\rho_{sh}$:

$$P\left(\|\epsilon_{sh}\|_2^2 \leq \sigma^2(2\ln(\sqrt{N}) + T_{sh} + 2\sqrt{T_{sh}\ln(\sqrt{N})})\right) > 1 - \frac{1}{\sqrt{N}}. \tag{10}$$

By choosing $\sigma$ less than $\frac{\rho_{sh}}{\sqrt{2\ln N^{1/2} + T_{sh} + 2\sqrt{T_{sh}\ln N^{1/2}}}}$ and $\min_i \frac{\rho_{ns}}{\sqrt{2\ln N^{1/2} + T_{i,ns} + 2\sqrt{T_{i,ns}\ln N^{1/2}}}}$, and referring to (9,10), we achieve both:

$$P\left(\|\epsilon_{ns}^i\| < \rho_{ns}\right) > 1 - \frac{1}{N^{1/2}}, \forall i,$$

$$P\left(\|\epsilon_{sh}\| < \rho_{sh}\right) > 1 - \frac{1}{N^{1/2}}.$$

Finally, we finish the proof as:

$$\left[\mathcal{L}_{\mathcal{D}}^i\left(\boldsymbol{\theta}^i\right)\right]_{i=1}^m \leq \mathbb{E}_{\epsilon_{sh}}\left[\mathbb{E}_{\epsilon_{ns}^i}\left[\mathcal{L}_{\mathcal{S}}^i\left(\boldsymbol{\theta}_{sh} + \epsilon_{sh}, \boldsymbol{\theta}_{ns}^i + \epsilon_{ns}^i\right)\right] + f^i\left(\|\boldsymbol{\theta}^i\|_2\right)\right]_{i=1}^m$$

$$\leq \max_{\|\epsilon_{sh}\| < \rho_{sh}}\left[\max_{\|\epsilon_{ns}^i\| < \rho_{ns}}\mathcal{L}_{\mathcal{S}}^i\left(\boldsymbol{\theta}_{sh} + \epsilon_{sh}, \boldsymbol{\theta}_{ns}^i + \epsilon_{ns}^i\right) + \frac{2}{\sqrt{N}} - \frac{1}{N} + f^i\left(\|\boldsymbol{\theta}^i\|_2\right)\right]_{i=1}^m$$

To reach the final conclusion, we redefine:

$$f^i\left(\|\boldsymbol{\theta}^i\|_2\right) = \frac{2}{\sqrt{N}} - \frac{1}{N} + f^i\left(\|\boldsymbol{\theta}^i\|_2\right).$$

Here we note that we reach the final inequality due to the following derivations:

$$\mathbb{E}_{\epsilon_{sh}}\left[\mathbb{E}_{\epsilon_{ns}^i}\left[\mathcal{L}_{\mathcal{S}}^i\left(\boldsymbol{\theta}_{sh} + \epsilon_{sh}, \boldsymbol{\theta}_{ns}^i + \epsilon_{ns}^i\right)\right]\right]_{i=1}^m$$

$$\leq \int_{B_{sh}}\left[\int_{B_{ns}^i}\mathcal{L}_{\mathcal{S}}^i\left(\boldsymbol{\theta}_{sh} + \epsilon_{sh}, \boldsymbol{\theta}_{ns}^i + \epsilon_{ns}^i\right)d\epsilon_{ns}^i + \frac{1}{\sqrt{N}}\right]_{i=1}^m d\epsilon_{sh}$$

$$+ \int_{B_{sh}^c}\left[\int_{B_{ns}^i}\mathcal{L}_{\mathcal{S}}^i\left(\boldsymbol{\theta}_{sh} + \epsilon_{sh}, \boldsymbol{\theta}_{ns}^i + \epsilon_{ns}^i\right)d\epsilon_{ns}^i + \frac{1}{\sqrt{N}}\right]_{i=1}^m d\epsilon_{sh}$$

$$\leq \int_{B_{sh}}\left[\int_{B_{ns}^i}\mathcal{L}_{\mathcal{S}}^i\left(\boldsymbol{\theta}_{sh} + \epsilon_{sh}, \boldsymbol{\theta}_{ns}^i + \epsilon_{ns}^i\right)d\epsilon_{ns}^i\right]_{i=1}^m d\epsilon_{sh} + \left(1 - \frac{1}{\sqrt{N}}\right)\frac{1}{\sqrt{N}} + \frac{1}{\sqrt{N}}$$

$$\leq \max_{\|\epsilon_{sh}\| < \rho_{sh}}\left[\max_{\|\epsilon_{ns}^i\| < \rho_{ns}}\left[\mathcal{L}_{\mathcal{S}}^i\left(\boldsymbol{\theta}_{sh} + \epsilon_{sh}, \boldsymbol{\theta}_{ns}^i + \epsilon_{ns}^i\right)\right]\right]_{i=1}^m + \frac{2}{\sqrt{N}} - \frac{1}{N},$$

where $B_{sh} = \{\epsilon_{sh} : \|\epsilon_{sh}\| \leq \rho_{sh}\}$, $B_{sh}^c$ is the compliment set, and $B_{ns}^i = \{\epsilon_{ns}^i : \|\epsilon_{ns}^i\| \leq \rho_{ns}\}$.

## C  GRADIENT AGGREGATION STRATEGIES OVERVIEW

This section details how the gradient_aggregate operation is defined according to recent gradient-based multi-task learning methods that we employed as baselines in the main paper, including MGDA (Sener & Koltun, 2018), PCGrad (Yu et al., 2020), CAGrad (Liu et al., 2021a) and IMTL (Liu et al., 2020). Assume that we are given $m$ vectors $\boldsymbol{g}^1, \boldsymbol{g}^2, \ldots, \boldsymbol{g}^m$ represent task gradients. Typically, we aim to find a combined gradient vector as:

$$\boldsymbol{g} = \text{gradient\_aggregate}(\boldsymbol{g}^1, \boldsymbol{g}^2, \ldots, \boldsymbol{g}^m)$$

.

## C.1 Multiple-gradient descent algorithm - MGDA

Sener & Koltun (2018) apply MGDA (Désidéri, 2012) to find the minimum-norm gradient vector that lies in the convex hull composed by task gradients $\boldsymbol{g}^1, \boldsymbol{g}^2, \ldots, \boldsymbol{g}^m$:

$$\boldsymbol{g} = \operatorname{argmin} ||\sum_{i=1}^{m} w_i \boldsymbol{g}^i||^2, s.t. \sum_{i=1}^{m} w_i = 1 \quad \text{and} \quad , w_i \geq 0 \forall i.$$

This approach can guarantee that the obtained solutions lie on the Pareto front of task objective functions.

## C.2 Projecting conflicting gradients - PCGrad

PCgrad resolves the disagreement between tasks by projecting gradients that conflict with each other, i.e. $\langle \boldsymbol{g}^i, \boldsymbol{g}^j \rangle < 0$, to the orthogonal direction of each other. Specifically, $\boldsymbol{g}^i$ is replaced by its projection on the normal plane of $\boldsymbol{g}^j$:

$$\boldsymbol{g}_{\text{PC}}^i = \boldsymbol{g}^i - \frac{\boldsymbol{g}^i \cdot \boldsymbol{g}^j}{||\boldsymbol{g}^j||^2} \boldsymbol{g}^j.$$

Then compute the aggregated gradient based on these deconflict vectors $\mathbf{g} = \sum_i^m \mathbf{g}_{\text{PC}}^i$.

## C.3 Conflict Averse Gradient Descent - CAGrad

CAGrad (Liu et al., 2021a) seeks a worst-case direction in a local ball around the average gradient of all tasks, $g_0$, that minimizes conflict with all of the gradients. The updated vector is obtained by optimizing the following problem:

$$\max_{\mathbf{g} \in R} \min_{i \in [m]} \langle \mathbf{g}^i, g \rangle \quad s.t. \quad ||\mathbf{g} - \mathbf{g}^0|| \leq c ||\mathbf{g}^0|,$$

where $\mathbf{g}^0 = \frac{1}{m} \sum_i^m \mathbf{g}^i$ is the averaged gradient and $c$ is a hyper-parameter.

## C.4 Impartial multi-task learning - IMTL

IMTL (Liu et al., 2020) proposes to balance per-task gradients by finding the combined vector $\mathbf{g}$, whose projections onto $\{\mathbf{g}^i\}_{i=1}^m$ are equal. Following this, they obtain the closed-form solution for the simplex vector $\boldsymbol{w}$ for reweighting task gradients:

$$\boldsymbol{w} = \boldsymbol{g}^1 \boldsymbol{U}^\top \left( \boldsymbol{D} \boldsymbol{U}^\top \right)^{-1}$$

where $\boldsymbol{u}^i = \boldsymbol{g}^i / \|\boldsymbol{g}^i\|$, $\boldsymbol{U} = [\boldsymbol{u}^1 - \boldsymbol{u}^2, \cdots, \boldsymbol{u}^1 - \boldsymbol{u}^m]$, and $\boldsymbol{D} = [\boldsymbol{g}^1 - \boldsymbol{g}^2, \cdots, \boldsymbol{g}^1 - \boldsymbol{g}^m]$ The aggregated vector is then calculated as $\mathbf{g} = \sum_i^m w_i \mathbf{g}^i$.

# D Implementation Details

In this part, we provide implementation details regarding the empirical evaluation in the main paper along with additional comparison experiments.

## D.1 Baselines

In this subsection, we briefly introduce some of the comparative methods that appeared in the main text:

- Linear scalarization (LS) minimizes the unweighted sum of task objectives $\sum_i^m \mathcal{L}^i(\boldsymbol{\theta})$.
- Scale-invariant (SI) aims toward obtaining similar convergent solutions even if losses are scaled with different coefficients via minimizing $\sum_i^m \log \mathcal{L}^i(\boldsymbol{\theta})$.
- Random loss weighting (RLW) (Lin et al., 2021) is a simple yet effective method for balancing task losses or gradients by random weights.

- Dynamic Weight Average (DWA) (Liu et al., 2019a) simply adjusts the weighting coefficients by taking the rate of change of loss for each task into account.
- GradDrop  (Chen et al., 2020) presents a probabilistic masking process that algorithmically eliminates all gradient values having the opposite sign w.r.t a predefined direction.

## D.2   Image classification

**Network Architectures.** For two datasets in this problem, Multi-MNIST and CelebA, we replicate experiments from (Sener & Koltun, 2018; Lin et al., 2019) by respectively using the Resnet18 (11M parameters) and Resnet50 (23M parameters) (He et al., 2016) with the last output layer removed as the shared encoders and constructing linear classifiers as the task-specific heads, i.e. 2 heads for Multi-MNIST and 40 for CelebA, respectively.

**Training Details.** We train the all the models under our proposed framework and baselines using:

- Multi-MNIST: Adam optimizer (Kingma & Ba, 2014) with a learning rate of $0.001$ for $200$ epochs using a batch size of $256$. Images from the three datasets are resized to $36 \times 36$.
- CelebA: Batch-size of $256$ and images are resized to $64 \times 64 \times 3$. Adam (Kingma & Ba, 2014) is used again with a learning rate of $0.0005$, which is decayed by $0.85$ for every $10$ epochs, our model is trained for $50$ epochs in total.

Regarding the hyperparameter for SAM (Foret et al., 2021), we use their adaptive version (Kwon et al., 2021) where both $\rho_{sh}$ and $\rho_{ns}$ are set equally and extensively tuned from $0.005$ to $5$.

## D.3   Scene understanding

Two datasets used in this problem are NYUv2 and CityScapes. Similar to (Navon et al., 2022), all images in the NYUv2 dataset are resized to $288 \times 384$ while all images in the CityScapes dataset are resized to $128 \times 256$ to speed up the training process. We follow the exact protocol in (Navon et al., 2022) for implementation. Specifically, SegNet (Badrinarayanan et al., 2017) is adopted as the architecture for the backbone and Multi-Task Attention Network MTAN (Liu et al., 2019a) is applied on top of it. We train each method for 200 epochs using Adam optimizer (Kingma & Ba, 2014) with an initial learning rate of $1e-4$ and reduced it to $5e-5$ after 100 epochs. We use a batch size of 2 for NYUv2 and 8 for CityScapes. The last 10 epochs are averaged to get the final results, and all experiments are run with three random seeds.

# E   Additional Results

To further show the improvement of our proposed training framework over the conventional one, this section provides additional comparison results in terms of qualitative results, predictive performance, convergent behavior, loss landscape, model sharpness, and gradient norm. Please note that in these experiments, we choose IMTL and F-IMTL as two examples for *standard* and *flat-aware* gradient-based MTL training respectively. We also complete the ablation study in the main paper by providing results on all three datasets in the Multi-MNIST dataset.

## E.1   Image segmentation qualitative result

In this section, we provide qualitative results of our method of the CityScapes experiment. We compare our proposed method against its main baseline by highlighting typical cases where our method excels in generalization performance. Figure 5 shows some visual examples of segmentation outputs on the test set. Note that in the CityScapes dataset, the "void" class is identified as unclear and pixels labeled as void do not contribute to either objective or score (Cordts et al., 2016).

While there is only a small gap between the segmentation performance of IMTL and F-IMTL, we found that a small area, which is the car hood and located at the bottom of images, is often incorrectly classified. For example, in Figure 5, the third and fourth rows compare the prediction of SegNet (Badrinarayanan et al., 2017) with ERM training and with our proposed method. It can be seen that both of them could not detect this area correctly, this is because this unclear "void" class did

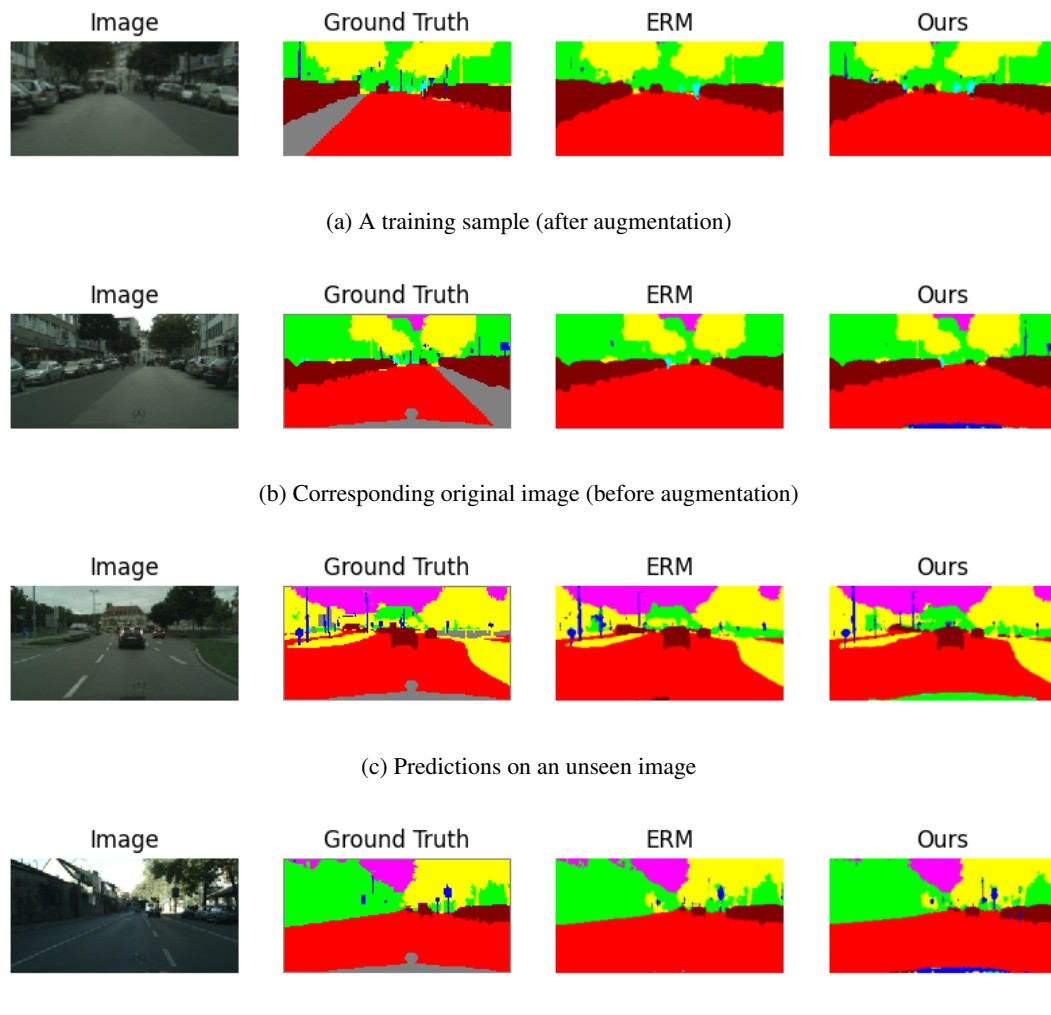

(a) A training sample (after augmentation)

(b) Corresponding original image (before augmentation)

(c) Predictions on an unseen image

(d) Predictions on an unseen image

Figure 5: Semantic segmentation prediction comparison on CityScapes . From left to right are input images, ground truth, and segmentation outputs from SegNet (Badrinarayanan et al., 2017) using ERM training and sharpness-aware training. Regions that are represented in gray color are ignored during training. (Best viewed in color).

not appear during training. Even worse, the currently employed data augmentation technique in the codebase of Nash-MTL and other recent multi-task learning methods Navon et al. (2022); Liu et al. (2021a) consists of RandomCrop, which often unintentionally excludes edge regions. For example, Figure 5a shows an example fed to the neural network for training, which excludes the car hood and its logo, compared to the original image (Figure 5b). Therefore, we can consider this "void" class as a novel class in this experiment, since its appearance is ignored in both training and evaluation. Even though, in Figures 5c and 5d our training method is still able to distinguish between this unknown area and other nearby known classes, which empirically shows the robustness and generalization ability of our method over ERM.

## E.2 PREDICTIVE PERFORMANCE

In this part, we provide experimental justification for an intriguing insight into the connection between model sharpness and model calibration. Empirically, we found that when a model converges to flatter minima, it tends to be more calibrated. We start by giving the formal definition of a well-calibrated

classification model and three metrics to measure the calibration of a model, then we analyze our empirical results.

Consider a $C$-class classification problem with a test set of $N$ samples given in the form $(x_i, y_i)_{i=1}^N$ where $y_i$ is the true label for the sample $x_i$. Model outputs the predicted probability for a given sample $x_i$ to fall into $C$ classes, is given by

$$\hat{\boldsymbol{p}}(x_i) = [\hat{p}(y = 1|x_i), \ldots, \hat{p}(y = C|x_i)].$$

$\hat{p}(y = c|x_i)$ is also the confidence of the model when assigning the sample $x_i$ to class $c$. The predicted label $\hat{y}_i$ is the class with the highest predicted value, $\hat{p}(x_i) := \max_c \hat{p}(y = c|x_i)$. We refer to $\hat{p}(x_i)$ as the confidence score of a sample $x_i$.

**Model calibration** is a desideratum of modern deep neural networks, which indicates that the predicted probability of a model should match its true probability. This means that the classification network should be not only accurate but also confident about its prediction, i.e. being aware of when it is likely to be incorrect. Formally stated, the *perfect calibration* (Guo et al., 2017) is:

$$P(\hat{y} = y|\hat{p} = q) = q, \forall q \in [0, 1]. \tag{11}$$

**Metric.** The exact computation of Equation 11 is infeasible, thus we need to define some metrics to evaluate how well-calibrated a model is.

- Brier score $\downarrow$ (BS) (Brier et al., 1950) assesses the accuracy of a model's predicted probability by taking into account the absolute difference between its confidence for a sample to fall into a class and the true label of that sample. Formally,

$$BS = \frac{1}{N} \sum_{i=1}^N \sum_{c=1}^C \left( \hat{p}(y = c|x_i) - \mathbf{1}[y_i = c] \right)^2.$$

- Expected calibration error $\downarrow$ (ECE) compares the predicted probability (or confidence) of a model to its accuracy (Naeini et al., 2015; Guo et al., 2017). To compute this error, we first bin the confidence interval $[0, 1]$ into $M$ equal bins, then categorize data samples into these bins according to their confidence scores. We finally compute the absolute value of the difference between the average confidence and the average accuracy within each bin, and report the average value over all bins as the ECE. Specifically, let $B_m$ denote the set of indices of samples having their confidence scores belonging to the $m^{th}$ bin. The average accuracy and the average confidence within this bin are:

$$acc(B_m) = \frac{1}{|B_m|} \sum_{i \in B_m} \mathbf{1}[\hat{y}_i = y_i],$$

$$conf(B_m) = \frac{1}{|B_m|} \sum_{i \in B_m} \hat{p}(x_i).$$

Then the ECE of the model is defined as:

$$ECE = \sum_{m=1}^M \frac{|B_m|}{N} |acc(B_m) - conf(B_m)|.$$

In short, the lower ECE neural networks obtain, the more calibrated they are.

- Predictive entropy (PE) is a widely-used measure of uncertainty (Ovadia et al., 2019; Lakshminarayanan et al., 2017; Malinin & Gales, 2018) via the predictive probability of the model output. When encountering an unseen sample, a well-calibrated model is expected to yield a high PE, representing its uncertainty in predicting out-of-domain (OOD) data.

$$PE = \frac{1}{C} \sum_{c=1}^C -\hat{p}(y = c|x_i) \log \hat{p}(y = c|x_i).$$

Figures 6 and 7 plot the distribution of the model's predicted entropy in the case of in-domain and out-domain testing, respectively. We can see when considering the flatness of minima, the model shows higher predictive entropy on both in-domain and out-of-domain, compared to ERM. This also means that our model outputs high uncertainty prediction when it is exposed to a sample from a different domain.

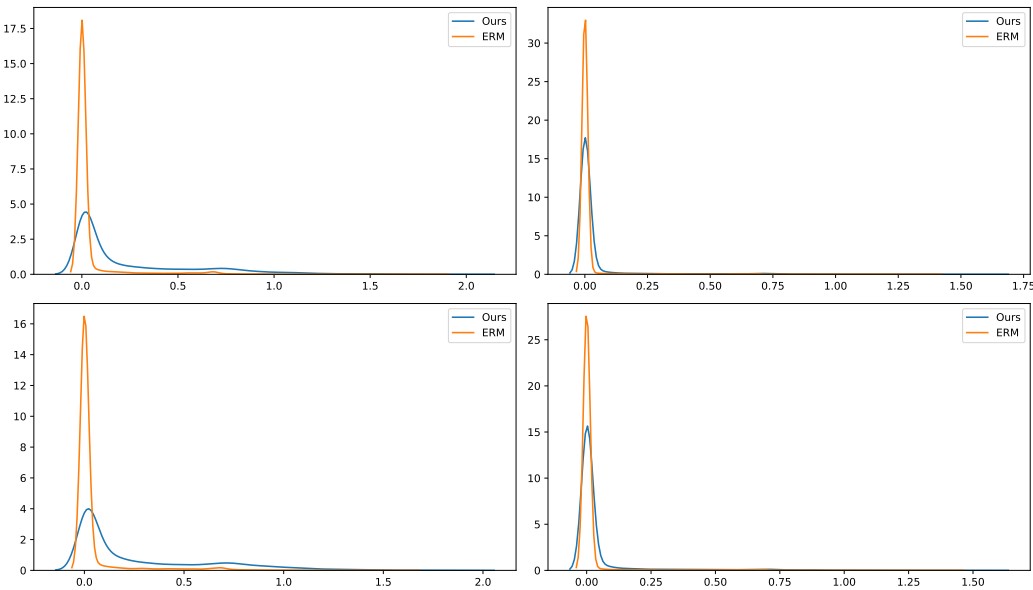

Figure 6: Histograms of predictive entropy of ResNet18 (He et al., 2016) on in domain dataset, train and test on MultiMNIST (left) and MultiFashion (right). We use the orange lines to denote ERM training while blue lines indicate our proposed method.

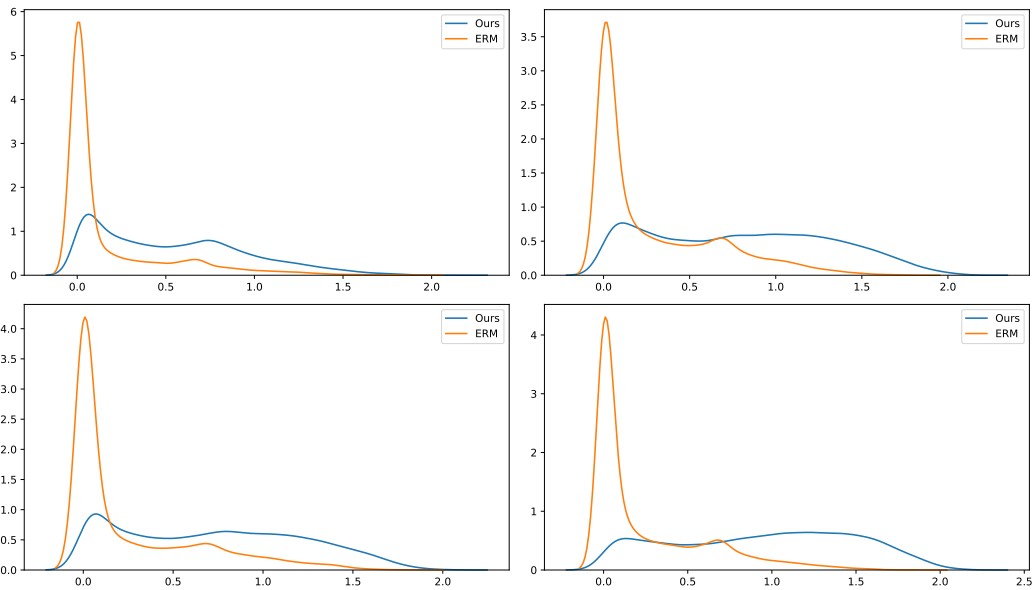

Figure 7: Out of domain: model is trained on MultiMNIST, then tested on MultiFashion (left) and vice versa (right). Models trained with ERM give over-confident predictions as their predictive entropy concentrates around 0.

Here, we calculate the results for both tasks 1 and 2 as a whole and plot their ECE in Figure 8. When we look at the in-domain prediction in more detail, our model still outperforms ERM in terms of expected calibration error. We hypothesize that considering flat minima optimizer not only lowers errors across tasks but also improves the predictive performance of the model.

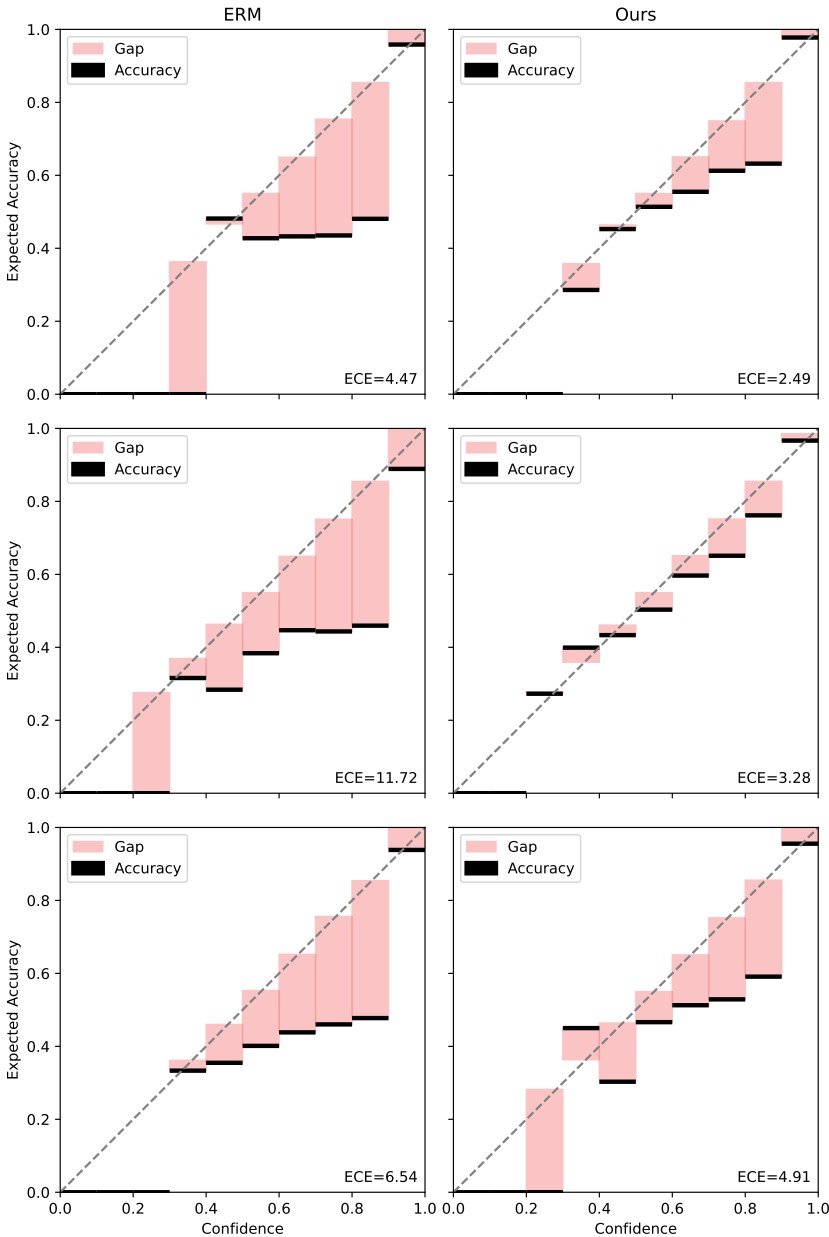

Figure 8: The predictive performance (measured by the expected calibration error) of neural networks has been enhanced by using our proposed training method (right column).

We also report the Brier score and ECE for each task in Table 6 and Table 7. As can be observed from these tables, our method shows consistent improvement in the model calibration when both scores decrease over all scenarios.

Table 6: Brier score on Multi-Fashion, Multi-Fashion+MNIST and MultiMNIST datasets. We use the **bold** font to highlight the best results.

| Dataset | Task | Multi-Fashion | Multi-Fashion+MNIST | MultiMNIST |
|---------|------|---------------|---------------------|------------|
| ERM | Top left | 0.237 | 0.055 | 0.082 |
| | Bottom right | 0.254 | 0.217 | 0.106 |
| | Average | 0.246 | 0.136 | 0.094 |
| Ours | Top left | **0.172** | **0.037** | **0.059** |
| | Bottom right | **0.186** | **0.189** | **0.075** |
| | Average | **0.179** | **0.113** | **0.067** |

Table 7: Expected calibration error on Multi-Fashion, Multi-Fashion+MNIST and MultiMNIST datasets. Here we set the number of bins equal to 10.

| Dataset | Task | Multi-Fashion | Multi-Fashion+MNIST | MultiMNIST |
|---------|------|---------------|---------------------|------------|
| ERM | Top left | 0.113 | 0.027 | 0.039 |
| | Bottom right | 0.121 | 0.104 | 0.050 |
| | Average | 0.117 | 0.066 | 0.045 |
| Ours | Top left | **0.034** | **0.015** | **0.022** |
| | Bottom right | **0.032** | **0.083** | **0.028** |
| | Average | **0.033** | **0.049** | **0.025** |

### E.3 EFFECT OF CHOOSING PERTURBATION RADIUS $\rho$.

The experimental results analyzing the sensitivity of model w.r.t $\rho$ are given in Figure 9. We evenly picked $\rho$ from 0 to 3.0 to run F-CAGrad on three Multi-MNIST datasets.

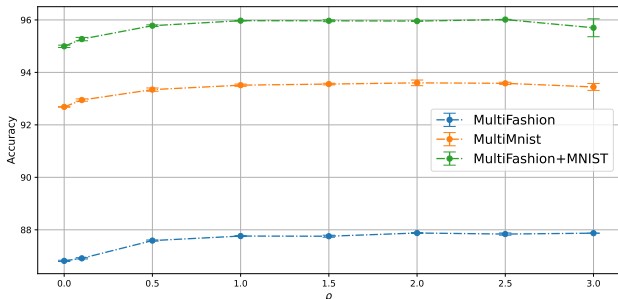

Figure 9: Average accuracy when varying $\rho$ from 0 to 3.0 (with error bar from three independent runs).

We find that the average accuracy of each task is rather stable from $\rho = 0.5$, which means the effect of different values of $\rho$ in a reasonably small range is similar. It can also easy to notice that the improvement tends to saturate when $\rho \geq 1.5$.

### E.4 GRADIENT CONFLICT.

In the main paper, we measure the percentage of gradient conflict on the MultiFashion+MNIST dataset. Here, we provide the full results on three different datasets. As can be seen from Figure 10, there is about half of the mini-batches lead to the conflict between task 1 and task 2 when using traditional training. Conversely, our proposed method significantly reduces such confliction (less than 5%) via updating the parameter toward flat regions.

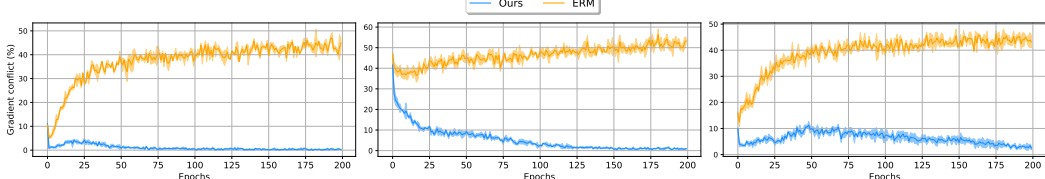

Figure 10: **Task gradient conflict proportion** of models trained with our proposed method and ERM across MultiFashion, MultiFashion+MNIST and MultiMNIST datasets (columns).

## E.5 LOSS LANDSCAPE

Thirdly, we provide additional visual comparisons of the loss landscapes trained with standard training and with our framework across two tasks of three datasets of Multi-MNIST. As parts of the obtained visualizations have been presented in the main paper, we provide the rest of them in this subsection. The results in Figure 11 consistently show that our method obtains significantly flatter minima on both two tasks, encouraging the model to generalize well.

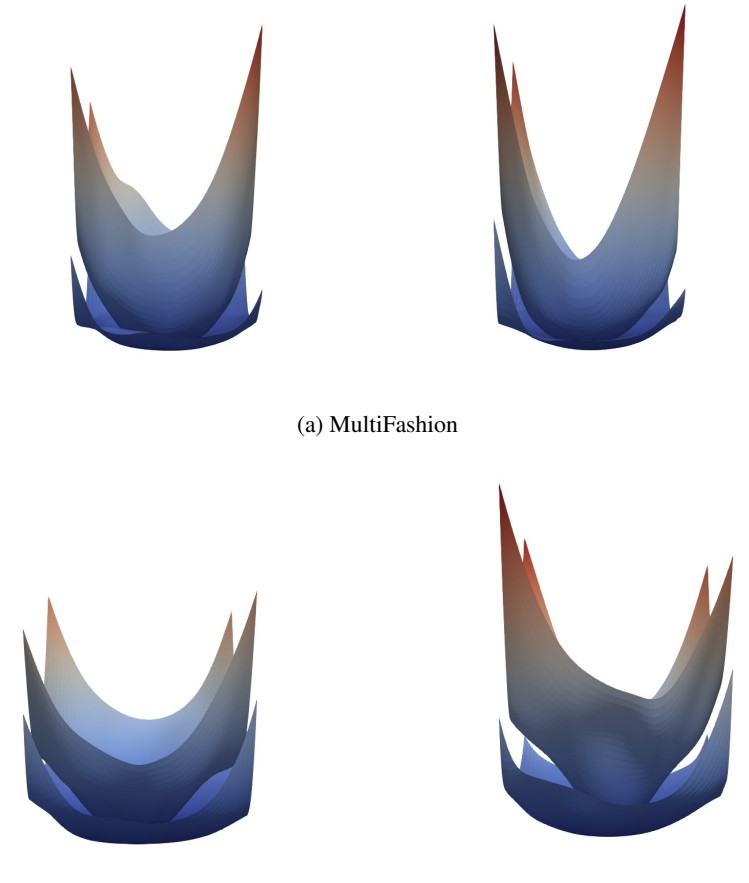

(a) MultiFashion

(b) MultiFashion+MNIST

Figure 11: Loss landscapes of task 1 and task 2 on MultiFashion and MultiFashion+MNIST, respectively.

## E.6 TRAINING CURVES

Secondly, we compare the test accuracy of trained models under the two settings in Fig. 13. It can be seen that from the early epochs (20-th epoch), the *flat-based* method outperforms the *ERM-based* method on all tasks and datasets. . Although the ERM training model is overfitted after such a long training, our model retains a high generalizability, as discussed throughout previous sections.

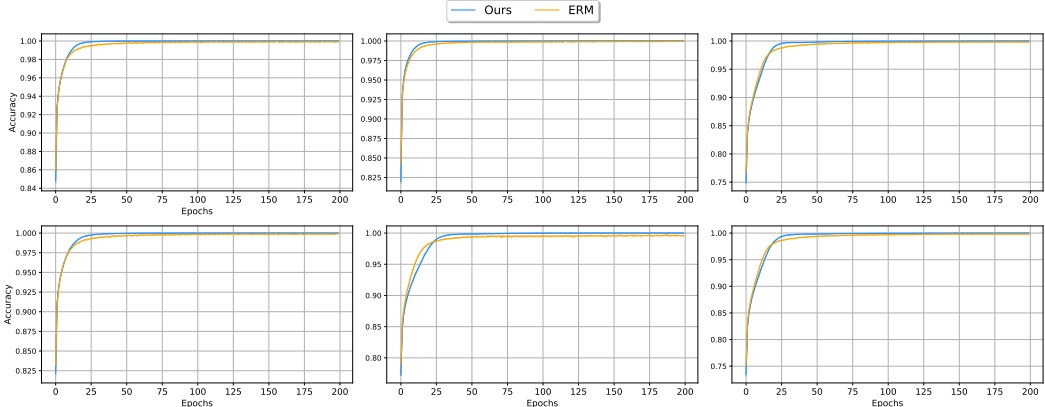

Figure 12: **Train accuracy** of models trained with our proposed method and ERM across 2 tasks (rows) of MultiFashion, MultiFashion+MNIST and MultiMNIST datasets (columns).

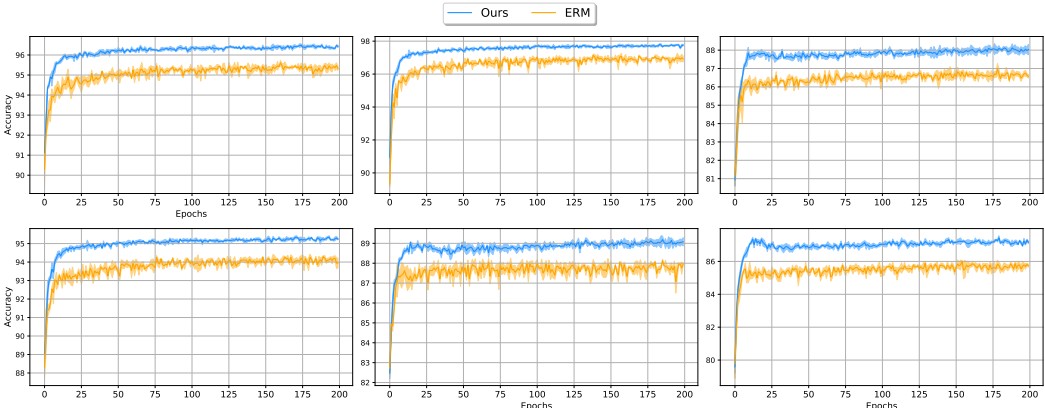

Figure 13: **Test accuracy** of models trained with our proposed method and ERM across 2 tasks (rows) of MultiFashion, MultiFashion+MNIST and MultiMNIST datasets (columns).

Furthermore, we also plot the training accuracy curves across experiments in Figure 12 to show that training accuracy scores of both ERM and our proposed method are similar and reach $\approx 100\%$ from 50-th epoch, which illustrates that the improvement is associated with generalization enhancement, not better training.

## E.7 MODEL SHARPNESS

Fourthly, Figure 14 displays the evolution of $\rho$-sharpness of models along training epochs under conventional loss function (ERM) and worst-case loss function (ours) on training sets of three datasets from Multi-MNIST, with multiple values of $\rho$. We can clearly see that under our framework, for both tasks, the model can guarantee uniformly low loss value in the $\rho$-ball neighborhood of parameter across training process. In contrast, ERM suffers from sharp minima from certain epochs when the model witnesses a large gap between the loss of worst-case perturbed model and current model. This is the evidence for the benefit that our framework brings to gradient-based methods, which is all tasks can concurrently find flat minima thus achieving better generalization.

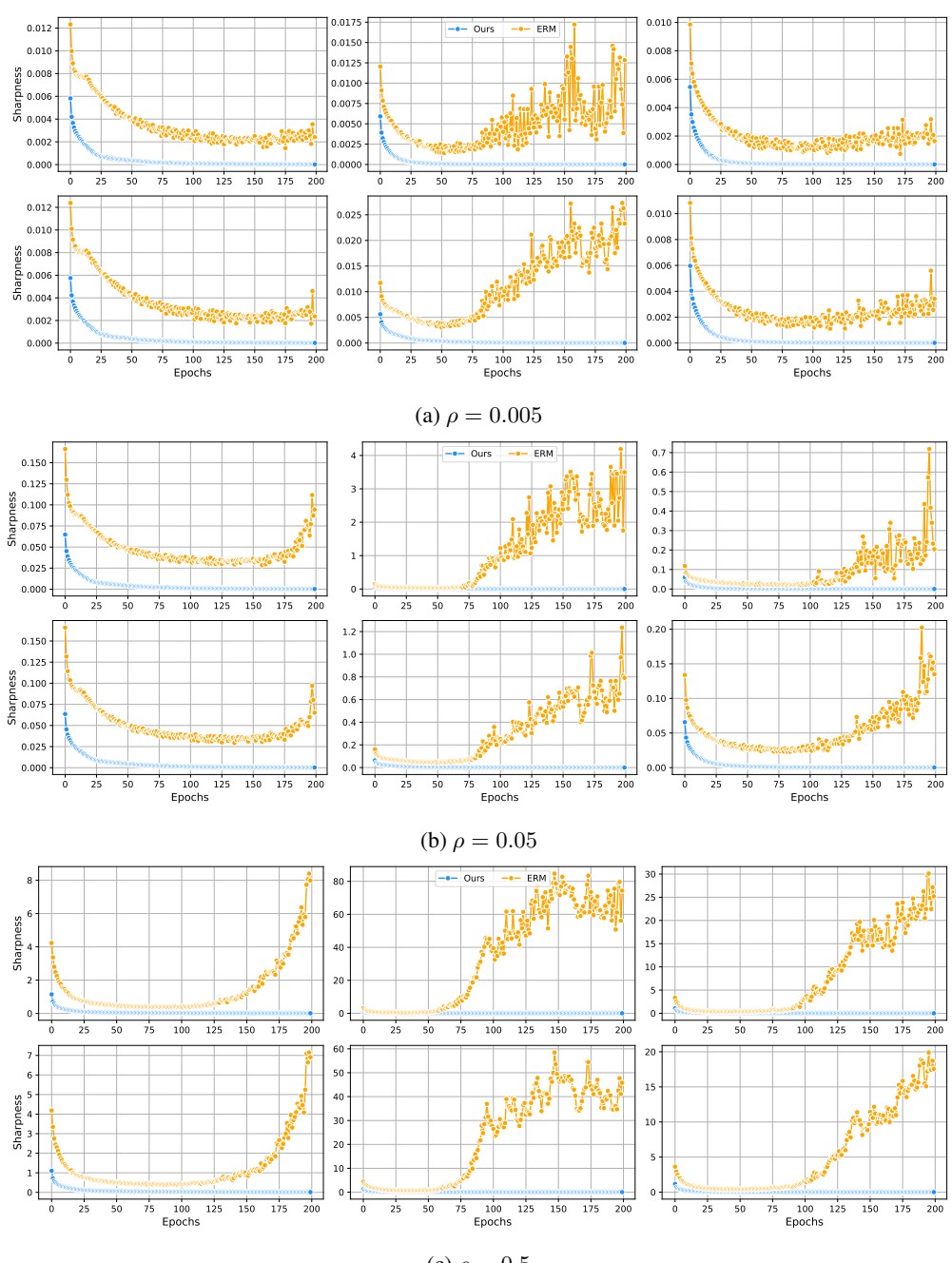

Figure 14: **Sharpness** of models trained with our proposed method and ERM with different values of $\rho$. For each $\rho$, the top and bottom row respectively represents the first and second task, and each column respectively represents each dataset in Multi-MNIST: from left to right are MultiFashion, MultiFashion+MNIST, MultiMNIST.

## E.8 GRADIENT NORM

Finally, we demonstrate the gradient norm of the loss function w.r.t the worst-case perturbed parameter of each task. On the implementation side, we calculate the magnitude of the flat gradient $g^{i,\text{flat}}$ for each task at different values of $\rho$ in Figure 15. As analyzed by equation (6) from the main paper, following the negative direction of $g_{sh}^{i,SAM}$ will lower the $\mathcal{L}_2$ norm of the gradient, which orients the

model towards flat regions. This is empirically verified in Figure 15. In contrast, as the number of epochs increases, gradnorm of the model trained with ERM tends to increase or fluctuate around a value higher than that of model trained with SAM.

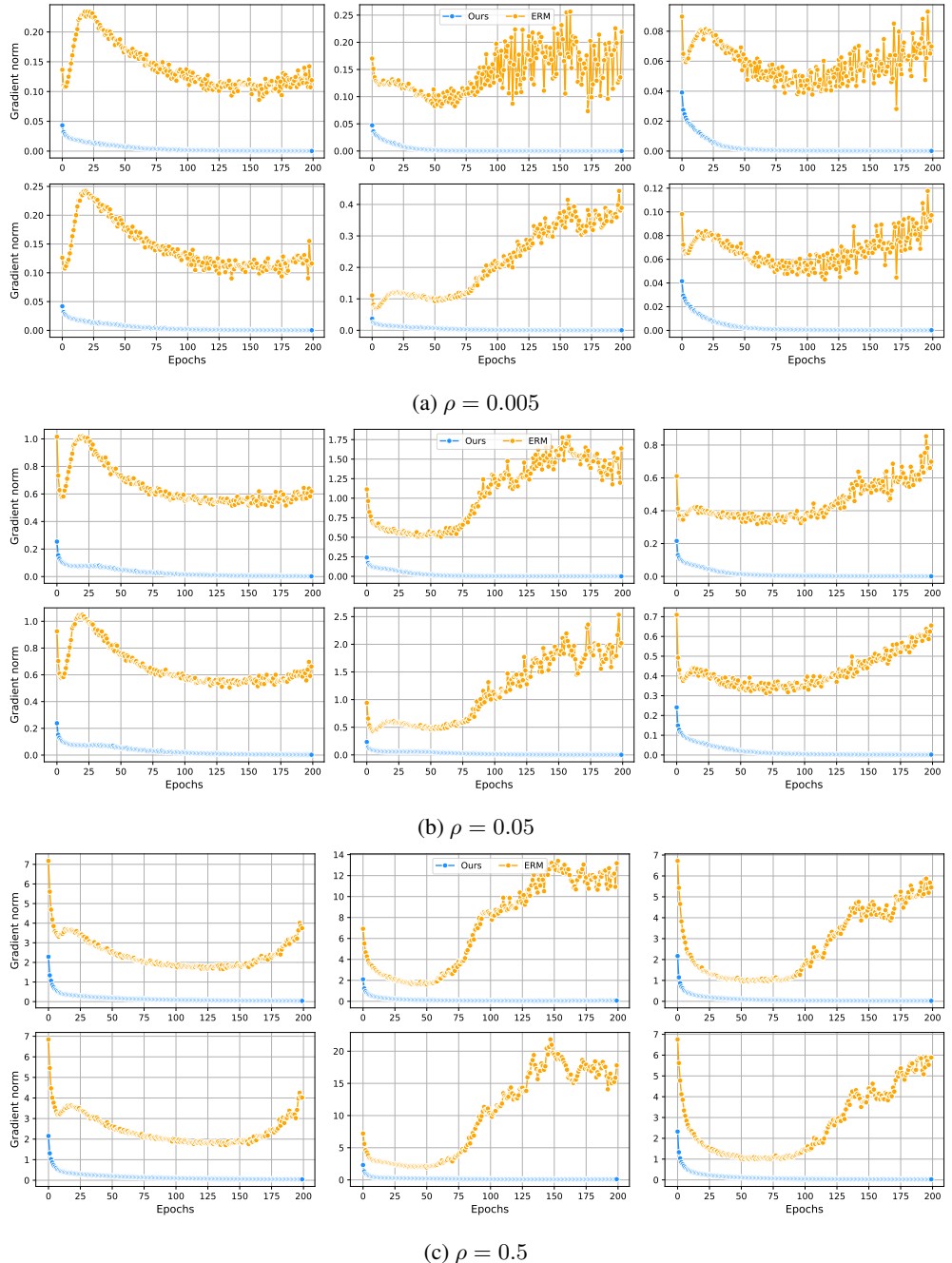

Figure 15: **Gradient magnitude** at the worst-case perturbations of models trained with our proposed method and ERM with different values of $\rho$. For each $\rho$, the top and bottom row respectively represents the first and second task, and each column respectively represents each dataset in Multi-MNIST: from left to right are MultiFashion, MultiFashion+MNIST, MultiMNIST.

