# OpenReview forum: "Improving Multi-task Learning via Seeking Task-based Flat Regions"
_ICLR.cc/2024/Conference — ICLR 2024 Conference Withdrawn Submission_

### Official Review · Reviewer_oZBs · 2023-10-13

**Soundness:** 2 fair
**Presentation:** 1 poor
**Contribution:** 2 fair
**Rating:** 3
**Confidence:** 5

**Summary:**

The paper applies sharpness-aware minimization (SAM) to multi-task learning (MTL), to find task-based flat minima for improving generalization capability on all tasks.  The paper conducts comprehensive experiments on several benchmark datasets to evaluate the proposed method.

**Strengths:**

- apply SAM to MTL is novel
- experimental results show that the proposed method can boost the performance of existing MTL methods on several benchmarks

**Weaknesses:**

- concerns about **efficiency**:
  - SAM is computationally expensive, doubling the computation cost compared with ERM/SGD. In Algorithm 1, each task requires computing the SAM gradient for shared/non-shared parameters. In total, the algorithm needs at least $2m$ gradient calculations, where $m$ is the number of tasks. Hence, the algorithm is computationally inefficient.
  - In experiments, there are no results (like training time) for comparing efficiency or training curve (performance w.r.t. training time).
  - this problem will be very serious when there are many tasks, e.g., the QM9 data set has 11 tasks.
  - some suggestions for mitigating this issue: use efficient variants of SAM, e.g.,
    - AE-SAM, An Adaptive Policy to Employ Sharpness-Aware Minimization, ICLR 2023
    - ESAM, Efficient Sharpness-aware Minimization for Improved Training of Neural Networks, ICLR 2022
- Eq(4) in Theorem 1, $[...]\_{i=1}^m \leq \max [...]\_{i=1}^m$ means ?
- Theorem 1 can be directly obtained from Theorem 1 of Foret et al. (2021): decomposing the parameters into two parts and using different $\rho$'s.
- in "Update the shared part" (P5), "However, a direct gradient aggregation  ... can be negatively affected by the gradient cancelation or conflict because it aims to combine many individual elements with different objectives", **a direct gradient aggregation means?** not clear
- Why the proposed aggregation in Section 4.4 is better than the above "direct gradient aggregation"?
- In the Conclusion Section, "proving that they can help enhance previous works both theoretically," which theorem(s)?
- how to calculate the entropy in Figure 2, note that the entropy in the figure has negative values.
- Figure 1, the  "2-task problem", where is the definition?

**Questions:**

see the questions in weakness part.

---

### Official Review · Reviewer_pVGE · 2023-10-30

**Soundness:** 3 good
**Presentation:** 3 good
**Contribution:** 2 fair
**Rating:** 3
**Confidence:** 4

**Summary:**

This paper presents a novel approach to Multi-Task Learning (MTL) by integrating Sharpness-aware Minimization, a technique that enhances single-task learning generalization. This new methodology aims to find flat minima for each task, improving overall generalization across multiple tasks. The paper showcases the effectiveness of this approach through extensive experiments, differentiating it from existing gradient-based MTL methods. The proposed method addresses the challenges of overfitting and negative transfer in MTL, contributing to more robust solutions in various applications.

**Strengths:**

The integration of SAM and MTL is somewhat new in transfer learning community. Furthermore, the application of SAM into existing gradient-based MTL studies is compatible. It improves the generalizability over various model architectures and tasks.

It is reasonable to assume that by leveraging flat minima on the multi-task learning setting, we could prevent over-fitting issue to the specific task or gradient intervention between different tasks.

The application of SAM on MTL requires some indirect adaptations. e.g. separate update rules for non-shared parts and shared part. The author successfully designed rules for each part.

**Weaknesses:**

The statement of Theorem 1 is too intuitive, which does not require rigorous proof on it. At the right side of Theorem 1, maximum of maximum is utilized for deriving the upper bound. It is intuitive based on my knowledge.

The analytical decomposition of SAM gradient into 1) loss and 2) flatness parts are not novel at all. It is well known analysis based on existing methods (SAM, GSAM, GAM). Rather, The new modeling parts of SAM-MTL is gradient decomposition on each task and gradient aggregation based on whole tasks. However, i do not get convinced why these gradient decomposition and re-organization are required in the context of multi-task learning. This is not empirically validated by additional ablation studies.

In Figure 4, the author claim that suggested algorithms significantly improves the task-wise flatness than ERM algorithm. What if we conduct simple SAM on MTL, not based on your gradient decomposition and re-organization? I conjecture that the flatness would be similar to SAM-MTL, your method. The extensive comparison with SAM variants (SAM,GSAM,GAM) is required.

Please empirically provide the computation cost increments by applying SAM-MTL. SAM is well known for increasing the computation cost about 2 times than ERM. is there any other increments during the adaptation of SAM-MTL?

**Questions:**

Please see Weaknesses section.

---

### Official Review · Reviewer_CubV · 2023-11-01

**Soundness:** 4 excellent
**Presentation:** 4 excellent
**Contribution:** 3 good
**Rating:** 8
**Confidence:** 3

**Summary:**

This work suggests a new framework to train multi-task learning (MTL) models that try to find a 'flat region' in the loss landscape. This is based on Sharpness-aware Minimization (SAM) by Foret et al. (2021), which was shown to reduce overfitting, and therefore could increase generalization performance across MTL tasks. The algorithm is based on solving a min-max optimization problem using Taylor expansion and gradient aggregation. Theorem establishes a generalization error bound. Experimental results on MTL computer vision tasks are provided.

**Strengths:**

This is a well-rounded paper. It's an extension of SAM by Foret et al. (2021), but the application of SAM to MTL is well-motivated. The algorithm is simple and easy to understand, and the derivation in sections 4.3-4.4 is clear. Authors present both theoretical and experimental analysis. I also appreciate that the authors uploaded their code for reproducibility, and provided detailed explanation for their experimental setup as well as interpretation of the results.

**Weaknesses:**

Please see questions below.

**Questions:**

1. The paper lacks a critical discussion on the limitations of this method. For example, is the method computationally efficient?

2. Are there standard deviations or statistical test results reported for Tables 2-4? It's not clear how significant some of these improvements are, e.g. 75.13 vs 75.77 in Table 3 PCGrad.

---

### Official Review · Reviewer_nSBj · 2023-11-07

**Soundness:** 2 fair
**Presentation:** 2 fair
**Contribution:** 2 fair
**Rating:** 3
**Confidence:** 4

**Summary:**

This paper combines sharpness-aware minimization (SAM) and existing gradient-based multitask learning algorithms to improve empirical generalization performance of MTL.  The main novelty is that the authors propose to decompose the SAM gradient $g^\textrm{SAM}$ into the task-loss minimizing direction, $g^\textrm{loss}$ (obtained by directly taking the directive w.r.t. task loss), and the flat-region seeking direction, $g^\textrm{flat}\coloneqq g^\textrm{SAM}-g^\textrm{loss}$, and perform gradient aggregation on both separately.  The proposed method, i.e., running existing gradient-based MTL algorithms by aggregating $g^\textrm{SAM}$ and $g^\textrm{loss}$ separately, is evaluated on a set of datasets, on average demonstrating improved performance v.s. just using $g^\textrm{loss}$ for parameter update.

**Strengths:**

1. The paper is well-motivated and presented.  Although I do find frequent grammatical errors, the paper is easy to read and understand.
2. It is an interesting observation that decomposing $g^\textrm{SAM}$ into and $g^\textrm{loss}$ and $g^\textrm{flat}$ and aggregating them separately is crucial for the success of the proposed method.  But this decomposition is—in the way it is currently presented—purely heuristic.  I would have liked more analyses on this beyond the ablation study on page 9.

**Weaknesses:**

1. Second point in strengths.

2. The proofs and theorems—which the authors claim to be a major contribution of the present work and on which the proposed algorithm is supposedly based—are poorly presented.  In turn, without which, the proposed approach is largely heuristic and lack theoretical support (excluding results that have been established in prior work, i.e., the constituent component of SAM and gradient-based MTL methods).

    - The "mild assumptions" are not clearly stated nor justified.  E.g., theorem 2 used the assumption that the loss function is bounded by $L$, which is not mentioned anywhere except in the proof.  Also, please justify and elaborate on the assumption that "that adding Gaussian perturbation will raise the test error": is it required for all $\theta$, or local minima?  It would be best if the assumptions are listed explicitly.

    - The conclusion of theorem 3 looks wrong.  First of all, in the proof, the induction is incorrectly applied—the $\xi$ cannot alter between cases.  The $\log1/\delta$ term in $f^i$ should be $\log m/\delta$.  And, does the conclusion not follow theorem 2 directly via a simple union bound?

    - The outer $\max _ {\\|\epsilon_\textrm{sh}\\|<\rho_\textrm{sh}}$ in the statement of Theorem 1 and 3 does not make sense to me.  The max is taken over a vector of $m$ dimensions.  ~~Is the max coordinate-wise?  If so, it should go inside the square bracket.  If not,~~ is the max well-defined?  Or, how is the total order of the vector space defined?

3. Regardless of the above potential issue with the theorem statement, I fail to see the connection between Theorem 1 (or its complete version 3) and the approach in section 4.3, i.e., the idea that "the worst-case shared perturbation $\epsilon_\mathrm{sh}$ is commonly learned for all tasks".  Specifically, how is computing the worst-case perturbation on each task separately and then aggregate the gradients $\\{g^{i,\textrm{SAM}}_\textrm{sh}\\} _ {i\in m}$ related to the idea above?

4. As mentioend in point 1 of strengths, there are some grammatical issues and weird word choice that may lead to confusions.  E.g., what is the "**ultimate** gradient descent direction" (in the abstract)?  Also, "is the compliment set" --> "is the complement set".

**Questions:**

See weaknesses.